# IRPM: Intergroup Relative Preference Modeling for Pointwise Generative Reward Models

Haonan Song [* 1]   Qingchen Xie [* 2]   Huan Zhu [* 1]   Feng Xiao [1]   Luxi Xing [1]   Liu Kang [1]   Fuzhen Li [1]
Zhiyong Zheng [1]   Feng Jiang [1]   Ziheng Li [1]   Kun Yan [3]   Qingyi Si [4]   Yanghua Xiao [5]   Hongcheng Guo [5]
Fan Yang [2]

## Abstract

Generative Reward Models (GRMs) have demonstrated strong performance in reward modeling, due to their interpretability and potential for refinement through reinforcement learning (RL). However, widely used pairwise GRMs create a computational bottleneck in reinforcement learning from human feedback (RLHF), when calibrating or aggregating preference signals over $n$ candidates, often incurring $\mathcal{O}(n^2)$ pairwise judgments. To address this issue, we propose **Intergroup Relative Preference Modeling (IRPM)**, an RL-based method that extends the Bradley–Terry preference-learning paradigm via intergroup comparisons to train *pointwise* GRMs from pairwise preference data. IRPM derives a pointwise reward for each response by contrasting groups of chosen vs. rejected samples, enabling pointwise scores comparable across candidate sets and $\mathcal{O}(n)$ reward evaluation for a variable number of candidates during RL training, while preserving interpretability and scalability. Experiments show that IRPM achieves state-of-the-art performance among pointwise GRMs on JudgeBench and RewardBench, and approaches the performance of leading pairwise GRMs. In addition, IRPM achieves substantial gains in post-training evaluations, demonstrating its effectiveness. Our models and code are available at https://github.com/XINGYUN-AI-LAB/IRPM.

[*]Equal contribution [1]HUJING Digital Media & Entertainment Group (XingYun Lab), Beijing, China [2]Department of Automation, Tsinghua University, Beijing, China [3]Beihang University, Beijing, China [4]Institute of Information Engineering, Chinese Academy of Sciences, Beijing, China [5]Fudan University, Shanghai, China. Correspondence to: Feng Xiao <ron.xf@alibaba-inc.com>, Hongcheng Guo <guohc@fudan.edu.cn>, Fan Yang <yangfan@tsinghua.edu.cn>.

*Proceedings of the $43^{rd}$ International Conference on Machine Learning*, Seoul, South Korea. PMLR 306, 2026. Copyright 2026 by the author(s).

## 1. Introduction

Large Language Models (LLMs) have achieved impressive performance across a wide range of applications (OpenAI et al., 2024; Guo et al., 2025). However, reliably aligning LLMs' behavior with human preferences remains a central challenge (Ouyang et al., 2022; Christiano et al., 2023). In modern alignment pipelines, reward models are a key component that provide reward signals for reinforcement learning from human feedback (RLHF). Explicit reward modeling approaches are categorized into two types: scalar reward models (Cobbe et al., 2021; Wang et al., 2024a;c; Liu et al., 2024) and generative reward models (GRMs) (Zhang et al., 2024; Guo et al., 2026).

In practical RLHF deployments, a reward model is often expected to satisfy four properties (illustrated in Figure 1): (1) flexibility: the method should accommodate different protocols with desirable performance (Li et al., 2023; Zhuang et al., 2024); (2) interpretability (Wang et al., 2024b; Zhang et al., 2025); (3) fine-grained reward capability: the method can produce calibrated per-response scores that remain comparable across candidate sets (Wu et al., 2023); and (4) inference-time scalability: the method can improve performance by allocating more compute at inference time, e.g., by allocating more computation to deliberative reasoning (Guo et al., 2026), or by sampling multiple judgments and aggregating them (Liu et al., 2025b). Under these criteria, existing reward modeling paradigms exhibit distinct trade-offs.

Scalar reward models are commonly trained with Bradley–Terry (B–T) loss (Bradley & Terry, 1952), which maximizes reward differences under pairwise comparisons. Because B–T produces only scalar scores and lacks explicit reasoning, it offers limited interpretability (Gao et al., 2023; Zhou et al., 2025). It also has limited inference-time scalability (Liu et al., 2025b), as summarized in Figure 1. These limitations motivate generative reward modeling approaches. Compared to scalar reward models, GRMs produce interpretable textual feedback, making them generally more interpretable and enabling inference-time scaling. Moreover, prior work has shown that GRMs trained on the same data can outper-

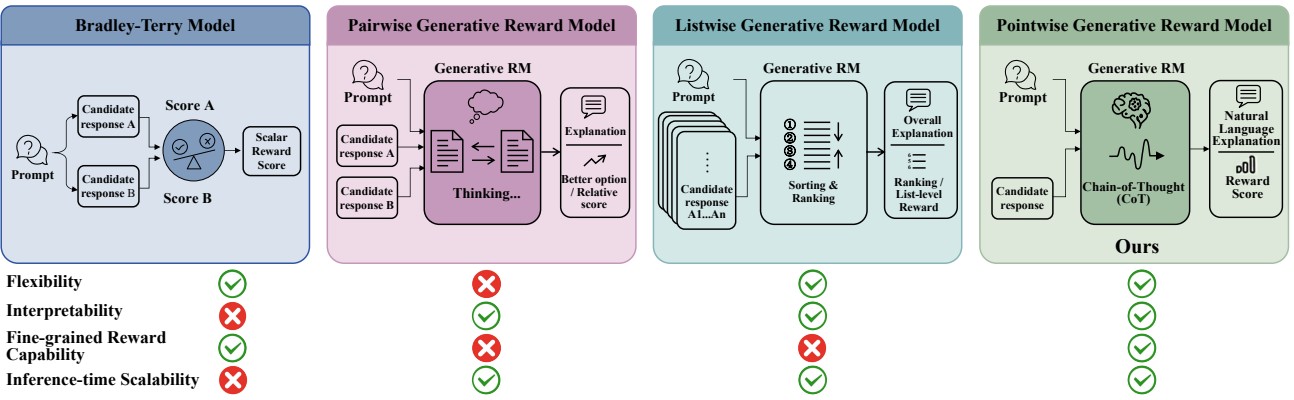

*Figure 1.* Different reward generation paradigms and scoring patterns for reward modeling, including the Bradley–Terry model, pairwise GRMs, listwise GRMs and pointwise GRMs. We compare these modeling approaches based on the following criteria: (1) flexibility: whether the method accommodates different protocols with desirable performance; (2) interpretability; (3) fine-grained reward capability: producing calibrated absolute scores that are comparable across candidate sets; and (4) inference-time scalability: the ability to improve performance by allocating more computation at inference time.

form B–T models (Mahan et al., 2024; Wang et al., 2025c;a). As shown in Figure 1, GRMs can be categorized into three types based on scoring patterns: pairwise (Chen et al., 2025; Whitehouse et al., 2025), listwise (Yu et al., 2025), and pointwise (Cao et al., 2024; Xu et al., 2025). Pairwise GRMs are widely adopted because they can capture fine-grained differences via pairwise comparisons and have achieved strong performance across multiple benchmarks. However, pairwise GRMs are less flexible in RL: to serve as a reward signal, relative preferences need to be converted into calibrated scalar scores. In practice, this often requires exhaustive all-pairs comparisons within each candidate set, incurring an $\mathcal{O}(n^2)$ cost for $n$ candidates, which is computationally expensive. In addition, this conversion can introduce a mismatch between evaluation and downstream usage, degrading practical effectiveness (Lambert & Calandra, 2024). Listwise GRMs evaluate a whole candidate set and output a ranking or scores, improving flexibility. But their scores can shift as the candidate set changes, making fine-grained reward challenging (Yan et al., 2022; Ren et al., 2024). Pointwise GRMs produce per-response absolute scores and satisfy all four desiderata. However, training pointwise GRMs typically relies on scalar annotations for individual responses, which are more costly to obtain at scale than pairwise preferences. We observe that the B–T model can leverage large-scale preference data to train pointwise models. This raises a natural question: *can we train a pointwise GRM using only pairwise preference data, so that we inherit the advantages of pointwise modeling while avoiding costly scalar annotations?*

To address this question, we propose **Intergroup Relative Preference Modeling (IRPM)**, an RL method that extends the B–T paradigm to train pointwise GRMs from pairwise

preference data. IRPM separately samples groups of rollouts for the chosen and rejected responses under the same prompt, performs *intergroup* comparisons to generate dense and stable rewards for each rollout, and then computes advantages by normalizing within-group rewards to update the model. Crucially, IRPM reduces reward evaluation during RL training to linear time complexity, $\mathcal{O}(n)$, substantially lowering computational cost compared with pairwise approaches that scale as $\mathcal{O}(n^2)$. By incorporating Chain-of-Thought (CoT) reasoning and adaptive preference strength, IRPM further improves performance. Experimental results show that IRPM outperforms prior state-of-the-art pointwise GRMs by an average of 2.8% on three benchmarks, while achieving performance competitive with leading pairwise GRMs. Finally, in post-training evaluations, IRPM outperforms pairwise GRMs at a substantially lower computational cost.

Our main contributions are as follows:

- We introduce IRPM, a method that satisfies the four key properties for reward models discussed above and achieves competitive performance across four benchmarks.

- We investigate practical strategies for improving GRMs via RL with CoT reasoning and adaptive preference strength.

- We show that IRPM achieves performance competitive with strong pairwise GRMs in post-training, while substantially reducing computational cost.

- We open-source the associated resources, including code and model checkpoints, to support reproducibility and enable further research.

**Conflict of Interest Disclosure.** Some authors are employees of Alibaba-affiliated companies. This work uses publicly available Qwen models developed by Alibaba as base models for reward modeling experiments.

## 2. Related Work

**Bradley–Terry Models** (Bradley & Terry, 1952) are commonly used to train scalar reward models from pairwise preference data and are widely adopted in RLHF. Our method adopts this paradigm and extends it to intergroup comparisons.

**Generative Reward Models.** Recent work has increasingly adopted LLM-as-a-judge style GRMs (Zheng et al., 2023). Pointwise GRMs such as Auto-J distill powerful judge models into a pointwise scoring model using supervised fine-tuning (SFT) (Li et al., 2023), and CompassJudger-1 explores combining pointwise and pairwise signals in a joint setting (Cao et al., 2024). Beyond SFT, several works show that RL training can improve reward modeling by leveraging reasoning: RM-R1 (Chen et al., 2025) and RRM (Guo et al., 2026) cast preference learning into an RLVR task, and J1 introduces a multitask RL framework with auxiliary pointwise objectives (Whitehouse et al., 2025). Concurrent work, PaTaRM (Jian et al., 2026), also studies training pointwise GRMs from pairwise data, but the core motivation differs. PaTaRM focuses on improving the alignment between pairwise supervision and pointwise inference. In contrast, our goal is to reduce the computational overhead of pairwise GRMs during RL training and to obtain a reward model that satisfies the four properties in Figure 1.

**Efficiency of Pairwise GRMs in RL Training.** A common way to integrate pairwise GRMs into RL is to compare candidates within each prompt and convert relative judgments into per-response rewards (Elo, 1978; Wang et al., 2025b), which naively requires all-pairs comparisons and incurs $\mathcal{O}(n^2)$ complexity. To mitigate this, RRM (Guo et al., 2026) uses a knockout tournament to reduce comparisons to roughly $\mathcal{O}(n \log n)$, while BRPO (Jia et al., 2025) further reduces the cost to $\mathcal{O}(n)$ by sampling a reference candidate for advantage estimation. However, the potential performance loss introduced by this approximation is not explicitly quantified.

**Group Relative Policy Optimization (GRPO)** (Shao et al., 2024) is a variant of PPO (Schulman et al., 2017) that obviates the need for additional value function approximation. For each prompt, GRPO performs a group rollout by sampling multiple candidate outputs from the current policy model, and uses the group's average reward as a baseline. The optimization objective is to increase the likelihood of outputs that outperform the baseline while penalizing those that underperform; details are provided in Appendix C. In-

spired by GRPO's rollout mechanism, we perform group rollouts for both the chosen and rejected responses and optimize the model using an intergroup reward signal.

## 3. Intergroup Relative Preference Modeling

This section introduces Intergroup Relative Preference Modeling (IRPM), a method that learns a pointwise GRM from pairwise preference data. Our key technical step is a theoretical derivation that reformulates the classical B–T pairwise preference likelihood under *stochastic* pointwise utilities produced by a GRM: the B–T preference probability becomes an expectation over score samples, which admits a tractable Monte Carlo estimator. Based on this Monte Carlo B–T form, we further derive a family of rollout-level intergroup training signals, including a soft probability decomposition, a hard AUC-style comparison, and several rule-based approximations. Finally, we optimize the GRM with a standard RL update rule (e.g., GRPO) using these rollout-level rewards. The overall pipeline is illustrated in Figure 2. We begin by formalizing the GRM's input and output in Section 3.1, then present the Monte Carlo B–T intergroup comparison objective in Section 3.2, design rollout-level intergroup rewards in Section 3.3, and describe the GRPO-based optimization procedure in Section 3.4.

### 3.1. Problem Setup

We consider reward modeling from pairwise preferences $\mathcal{D} = \{(x^{(i)}, y_c^{(i)}, y_r^{(i)})\}_{i=1}^N$, where $x$ is a prompt and $(y_c, y_r)$ denotes a chosen–rejected response pair. Our goal is to learn a pointwise GRM that, given a prompt-response pair $(x, y)$, generates (i) a natural-language critique $c$ and (ii) a scalar score $s$. The prompt template is provided in Appendix A.1. We model the GRM as an autoregressive generator with the following factorization:

$$p_\theta(c, s \mid x, y) = p_\theta(c \mid x, y)\, p_\theta(s \mid x, y, c). \quad (1)$$

### 3.2. From B–T to Rollout-Based Intergroup Comparisons

**B–T models.** Given a deterministic pointwise utility function $r_\theta(x, y)$, the B–T model defines the probability that $y_c$ is preferred to $y_r$ as

$$p_\theta(y_c \succ y_r \mid x) = \sigma\big(r_\theta(x, y_c) - r_\theta(x, y_r)\big), \quad (2)$$

and learns $r_\theta$ by maximizing the log-likelihood of pairwise preferences.

**Stochastic utilities induced by a GRM.** In our setting, the GRM produces a *stochastic* scalar score for a fixed $(x, y)$ by sampling $(c, s) \sim p_\theta(c, s \mid x, y)$. We interpret the generated scalar $s$ as a random utility sample, so the B–T

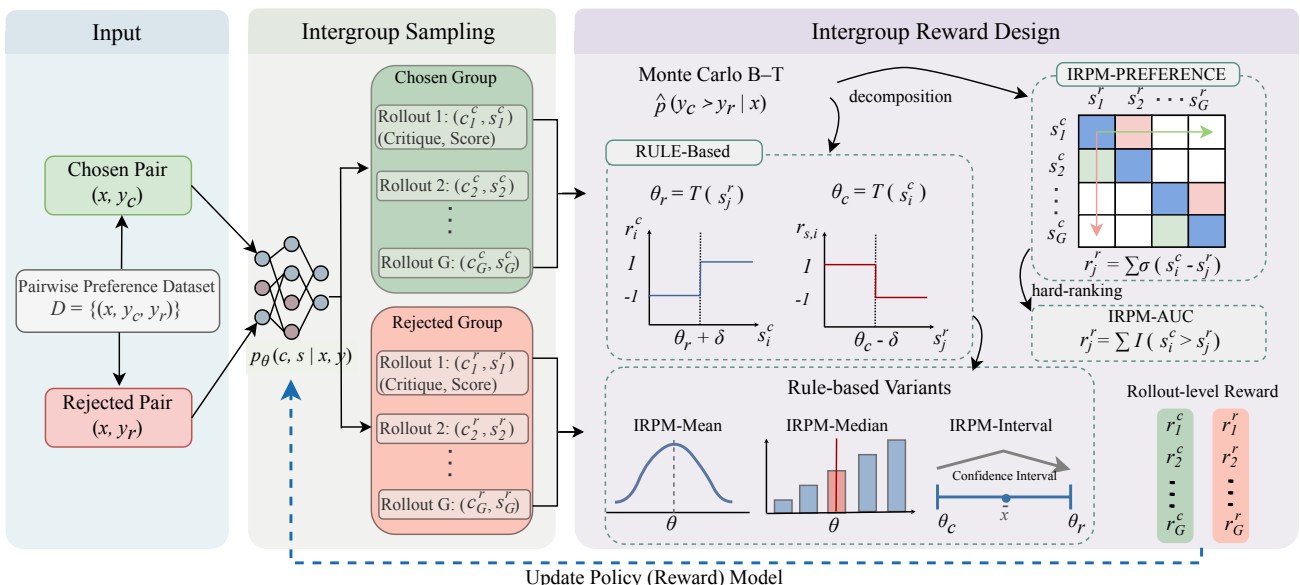

*Figure 2.* Overview of the RL pipeline of IRPM. Given a pairwise preference dataset $\mathcal{D} = \{(x^{(i)}, y_c^{(i)}, y_r^{(i)})\}_{i=1}^N$, IRPM takes a single prompt–response pair $(x, y)$ as input and produces a critique $c$ followed by a scalar score $s$. For each chosen/rejected pair, IRPM samples $G$ rollouts with the GRPO rollout mechanism to form two completion groups, and computes intergroup rewards through intergroup comparisons: including IRPM-Preference, IRPM-AUC and rule-based variants, yielding rollout-level rewards. Finally, the policy is optimized using the same objective and update procedure as GRPO.

preference probability can be rewritten as an expectation over two independent score samples:

$$p_\theta(y_c \succ y_r \mid x) \triangleq \mathbb{E}_{\substack{s^c \sim p_\theta(\cdot \mid x, y_c) \\ s^r \sim p_\theta(\cdot \mid x, y_r)}} \left[ \sigma(s^c - s^r) \right]. \quad (3)$$

**Monte Carlo B–T estimator via intergroup sampling.**
For each labeled pair $(x, y_c, y_r)$, we draw $G$ i.i.d. score samples from each side to form a chosen group $\{s_i^c\}_{i=1}^G$ and a rejected group $\{s_j^r\}_{j=1}^G$. The expectation in Equation (3) then admits a straightforward Monte Carlo estimate based on $G \times G$ intergroup comparisons:

$$\hat{p}(y_c \succ y_r \mid x) = \frac{1}{G^2} \sum_{i=1}^G \sum_{j=1}^G \sigma(s_i^c - s_j^r). \quad (4)$$

Moreover, the Monte Carlo form in Equation (4) naturally induces rollout-level training signals by assigning each sampled score a reward through comparisons against the opposite group.

### 3.3. Intergroup Reward Design

Starting from the Monte Carlo B–T estimator in Equation (4), we derive several rollout-level intergroup rewards. These designs differ in how they instantiate the intergroup comparison kernel (soft vs. hard) or approximate the full

$G \times G$ comparisons with simpler statistics, while remaining grounded in the same Monte Carlo reformulation. We evaluate all methods in Section 4.3.

**IRPM-Preference (Monte Carlo B–T decomposition).**
Because Equation (4) is a double average over intergroup comparisons, it can be exactly decomposed into per-rollout rewards by re-ordering the summations:

$$\hat{p}(y_c \succ y_r \mid x) = \frac{1}{G} \sum_{i=1}^G \underbrace{\left( \frac{1}{G} \sum_{j=1}^G \sigma(s_i^c - s_j^r) \right)}_{r_i^c}$$

$$= \frac{1}{G} \sum_{j=1}^G \underbrace{\left( \frac{1}{G} \sum_{i=1}^G \sigma(s_i^c - s_j^r) \right)}_{r_j^r}. \quad (5)$$

where $r_i^c$ measures how strongly the i-th chosen completion outranks rejected completions in expectation (and $r_j^r$ is defined analogously). Averaging these rollout-level rewards over the group recovers the same intergroup preference estimate in Equation (4).

**IRPM-AUC.** As a hard-ranking variant of the same intergroup Monte Carlo comparisons, we replace the soft kernel $\sigma(\cdot)$ with an indicator, yielding an AUC-style win-rate esti-

mator:

$$r_i^c = \frac{1}{G}\sum_{j=1}^{G}\mathbb{I}(s_i^c > s_j^r), \quad r_j^r = \frac{1}{G}\sum_{i=1}^{G}\mathbb{I}(s_i^c > s_j^r). \quad (6)$$

**Rule-based rewards.** As a further approximation to the full $G \times G$ Monte Carlo comparisons, we compare each rollout against a representative score level of the opponent group, summarized by an estimator $\hat{\theta}$, which yields a separable binary reward:

$$r_i^c = 2\mathbb{I}(s_i^c > \hat{\theta}_r + \delta) - 1, \quad r_j^r = 2\mathbb{I}(s_j^r < \hat{\theta}_c - \delta) - 1, \quad (7)$$

where $\mathbb{I}$ is an indicator and $\delta \geq 0$ is a tunable margin. We consider three choices of the estimator $\hat{\theta}$:

**IRPM-Mean.** Use the sample mean:

$$\hat{\theta} = \frac{1}{G}\sum_{i=1}^{G}s_i. \quad (8)$$

**IRPM-Median.** Use the sample median for robustness to outliers. Sort scores $s_1 \leq s_2 \leq \cdots \leq s_G$ and define:

$$\hat{\theta} = s_{\lceil 0.5G \rceil}. \quad (9)$$

**IRPM-Interval.** Use a confidence bound CI to account for sampling uncertainty. Details are provided in Appendix E. We adopt conservative thresholds:

$$\hat{\theta}_c := \mathrm{CI}_{\mathrm{lower}}^{(c)}, \qquad \hat{\theta}_r := \mathrm{CI}_{\mathrm{upper}}^{(r)}, \quad (10)$$

so that a rollout is rewarded only if the margin separation holds even under uncertainty.

**Format reward.** We additionally penalize completions that violate the required output format:

$$r_{\mathrm{format}} = \begin{cases} 0, & \text{if the output matches the format,} \\ -0.5, & \text{otherwise.} \end{cases}$$

$$(11)$$

The total reward is

$$r_{\mathrm{total}} = r_{\mathrm{intergroup}} + r_{\mathrm{format}}. \quad (12)$$

### 3.4. Updating the Policy Model

After computing intergroup rewards for all rollouts, we follow GRPO to compute *within-group* advantages and update the policy. Concretely, rewards are normalized *separately* within the chosen group and within the rejected group to form advantage estimates, and the policy is optimized using the standard GRPO objective, which encourages higher-probability generation of completions with higher relative advantages while maintaining training stability.

## 4. Experiments

### 4.1. Experimental Setup

**Training Dataset.** We use HelpSteer3-Preference (Wang et al., 2025c), a high-quality and diverse dataset of human-annotated preferences designed for training general-domain instruction-following language models via RLHF. It contains over 40,000 samples spanning a wide range of real-world scenarios, including STEM, coding, and multilingual tasks. Each example is annotated with a four-level preference score: 0–1 (*slightly better*), 2 (*better*), and 3 (*much better*). For preprocessing, we apply two filters: (1) to improve training performance, we exclude samples with low preference scores (0 or 1); and (2) to reduce training cost, we discard samples whose input length exceeds 3,072 tokens. After filtering, we retain approximately 21,000 samples.

**Benchmarks.** We evaluate IRPM on four benchmarks for reward models: RM-Bench, JudgeBench, RewardBench, and PPE. These benchmarks cover multilingual instructions and responses and include both verifiable and non-verifiable tasks. RM-Bench (Liu et al., 2025a) contains 4,000 samples to evaluate the robustness of reward models, focusing on sensitivity to subtle content changes and resistance to stylistic bias. JudgeBench (Tan et al., 2025) includes 350 challenging preference pairs spanning knowledge, reasoning, math, and coding. RewardBench (Lambert et al., 2025) is an early benchmark for evaluating reward models, comprising 3,000 samples across four categories: chat, chat-hard, reasoning, and safety. Preference Proxy Evaluations (PPE) (Frick et al., 2025) is a large-scale benchmark that links reward models to real-world human preference performance. It comprises two subsets: (i) PPE Preference, which includes 10,200 human preference pairs from Chatbot Arena; and (ii) PPE Correctness, which contains 12,700 response pairs evaluated on verifiable benchmarks (MMLU-Pro, MATH, GPQA, MBPP-Plus, and IFEval).

**Baselines.** We compare IRPM with reward models from three categories: (1) scalar reward models, including Armo-RM-8B (Wang et al., 2024b), Skywork-Reward-Gemma-2-27B (Liu et al., 2024), and DeepSeek-BTRM-27B (Liu et al., 2025b); (2) pairwise GRMs, including DeepSeek-GRM-27B (Liu et al., 2025b), EvalPlanner (Saha et al., 2025), RM-R1 (Chen et al., 2025), J1 (Whitehouse et al., 2025), and RRM (Guo et al., 2026); and (3) pointwise GRMs, including TIR (Xu et al., 2025) and PaTaRM (Jian et al., 2026).

**Implementation Details.** Unlike conventional SFT, which relies on curated reasoning traces, IRPM encourages the model to iteratively refine and expand its reasoning abilities via RL. We use Qwen3 (Yang et al., 2025) and Llama-3.1-Instruct (Grattafiori et al., 2024) as backbone models and conduct training using VERL (Sheng et al., 2025). We train

*Table 1.* Results on four benchmarks (RM-Bench, JGBench, RWBench and PPE). **Overall** denotes the average performance across the four benchmarks. The number after "+" indicates the absolute improvement over the corresponding backbone model of the same size. Bold marks the best overall results, and underlining indicates the best results among pointwise GRMs. We additionally report results with the Llama-3.1-8B-Instruct backbone and report results for the latest available version of PaTaRM.

| Models | Train Data | RM-Bench | JGBench | RWBench | PPE | | | Overall |
| --- | --- | --- | --- | --- | --- | --- | --- | --- |
| | | | | | Pref. | Corr. | Avg. | |
| **SOTA GRMs (Pairwise)** | | | | | | | | |
| DeepSeek-GRM-27B | 237k | — | — | 86.0 | 64.7 | 59.8 | 62.3 | — |
| EvalPlanner-Llama-8B | 22k | 68.1 | 30.2 | 83.0 | — | — | 54.3 | — |
| EvalPlanner-Llama-70B | 22k | 82.1 | 56.6 | **93.8** | 65.6 | 70.2 | 67.9 | 75.1 |
| RRM-7B | 420k | 70.4 | 67.0 | 82.2 | — | 65.4 | — | — |
| RRM-32B | 420k | 85.4 | 76.0 | 91.2 | — | 75.3 | — | — |
| RM-R1-DeepSeek-Distill-7B | 73k | 72.4 | 67.7 | 80.1 | — | 66.9 | — | — |
| RM-R1-DeepSeek-Distill-32B | 73k | 83.9 | **78.4** | 90.9 | — | 75.6 | — | — |
| J1-Llama-8B | 22k | 73.4 | 42.0 | 85.7 | — | — | 59.8 | — |
| J1-Qwen-32B-MultiTask | 22k | **90.3** | 71.4 | 93.6 | 66.8 | 76.8 | **71.8** | 81.8 |
| **Scalar Reward Models (Pointwise)** | | | | | | | | |
| Armo-RM-8B | 1000k | 67.7 | — | 90.3 | — | 61.2 | — | — |
| Skywork-Reward-Gemma-2-27B | 80k | 67.3 | — | **93.8** | 56.6 | 54.7 | 55.7 | — |
| DeepSeek-BTRM-27B | 237k | — | — | 81.7 | 68.3 | 66.7 | 67.5 | — |
| **Backbone (Pointwise)** | | | | | | | | |
| Llama-3.1-8B-Instruct | — | 39.7 | 34.6 | 50.7 | 32.8 | 37.7 | 35.3 | 40.1 |
| Qwen3-8B (thinking) | — | 73.1 | 62.0 | 80.5 | 58.3 | 62.5 | 60.4 | 69.0 |
| Qwen3-14B (thinking) | — | 77.3 | 66.9 | 83.1 | 60.5 | 66.1 | 63.3 | 72.7 |
| Qwen3-32B (thinking) | — | 76.8 | 68.9 | 85.2 | 60.4 | 67.9 | 64.2 | 73.8 |
| **SOTA GRMs (Pointwise)** | | | | | | | | |
| PaTaRM-Qwen3-8B | 38k | 78.7 | — | 84.3 | — | — | — | — |
| PaTaRM-Qwen3-14B | 38k | 80.3 | — | 87.2 | — | — | — | — |
| TIR-Judge-Zero-8B (Tool) | 26k | 76.3 | 67.5 | 81.4 | — | 70.3 | — | — |
| **Our Models (Pointwise)** | | | | | | | | |
| IRPM-Mean-Llama-8B | 21k | 65.7 +26 | 57.7 +23.1 | 78.1 +27.4 | 60.2 | 54.9 | 57.6 +22.3 | 64.8 +24.7 |
| IRPM-Mean-8B | 21k | 77.9 +4.8 | 70.0 +8.0 | 85.7 +5.2 | 62.8 | 68.7 | 65.8 +5.4 | 74.9 +5.9 |
| IRPM-Mean-14B | 21k | 80.3 +3.0 | 72.4 +5.5 | 86.3 +3.2 | 64.6 | 70.7 | 67.7 +4.4 | 76.7 +4.0 |
| IRPM-Mean-32B | 21k | 79.6 +2.8 | 74.6 +5.7 | 87.3 +2.1 | 64.3 | 70.0 | 67.2 +3.0 | 77.2 +3.4 |
| w/ voting@2 | 21k | 84.0 | 75.1 | 89.3 | 65.6 | 74.5 | 70.1 | 79.6 |
| w/ voting@8 | 21k | 84.8 | 77.7 | 91.2 | 66.1 | 77.3 | 71.7 | 81.4 |

with batch size 96, learning rate 5e-6, mini-batch size 96, number of rollouts 4, temperature of rollout 1.0, for 2 epochs. We save a checkpoint every 100 steps for performance evaluation. Details are provided in Appendix B.1. For Qwen3-32B, training required a total of 656 AMD MI300X GPU hours.

## 4.2. Results

We compare our model with established baselines using performance metrics reported in the corresponding publications and official leaderboards. Our model adopts the IRPM-Mean ($\delta$ is discussed in Section 4.3) described in Section 3.3, which achieves slightly better average performance than other reward design methods. The Qwen3 baseline is evaluated with the same prompts and protocol used for IRPM. The comprehensive results are summarized in Ta-

ble 1. Our observations and conclusions are as follows. All improvements mentioned in the comparison of experimental results refer to absolute gains.

**IRPM achieves state-of-the-art performance among pointwise GRMs.** Across all model sizes (8B, 14B, and 32B), IRPM delivers consistent improvements over the backbone across four benchmarks. Among the three model sizes, IRPM-Mean-8B achieves the largest average gain of 5.9% and shows strong training stability and generalization. Details are provided in Appendix D.1. IRPM-Mean-32B improves by 3.4% on average, exceeding the strongest pairwise baseline in our evaluation, J1-Qwen-32B-MultiTask (Whitehouse et al., 2025), which improves by 2.6% over its pairwise baseline on the same backbone and benchmarks. At the 8B-scale, IRPM-Mean-8B achieves the best performance on JudgeBench and RewardBench among pointwise GRMs. Compared with the tool-augmented pointwise baseline TIR-

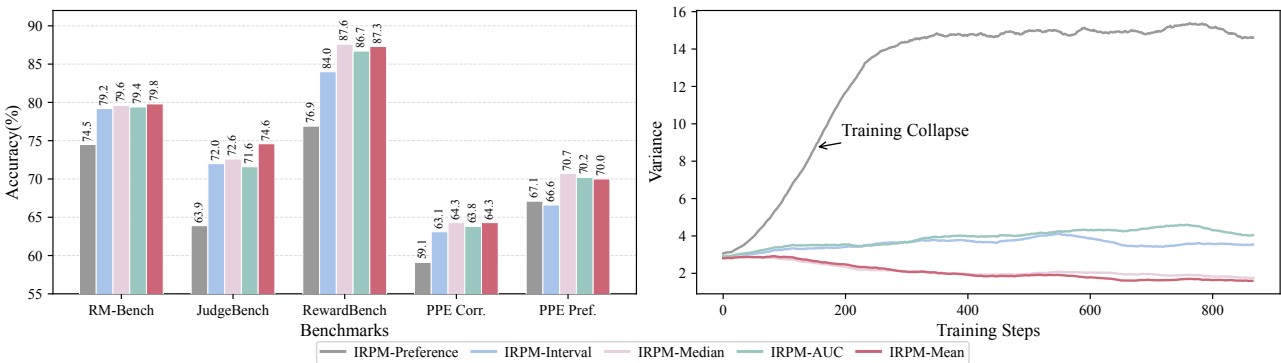

*Figure 3.* **Left:** Accuracy (%) of different reward design methods on RM-Bench, JudgeBench, RewardBench, and PPE (Preference and Correctness). **Right:** Per-batch variance of the training scores over training steps for each reward design method.

Judge-Zero-8B (Xu et al., 2025), IRPM-Mean-8B achieves higher scores on three benchmarks, with an average improvement of 2.8%. Notably, TIR relies on external code execution, which has been shown to improve performance. IRPM-Mean-8B also performs comparably to the pairwise GRM EvalPlanner-Llama-70B (Saha et al., 2025) across the four benchmarks. Because prior work reports relatively few pointwise baselines, we include strong pairwise GRMs as reference points. Although IRPM-Mean-32B trails the best pairwise GRMs on average, it matches or surpasses them on several benchmarks: it outperforms DeepSeek-GRM-27B (Liu et al., 2025b) on PPE, and surpasses EvalPlanner-Llama-70B and J1-Qwen-32B-MultiTask on JudgeBench. Furthermore, IRPM-Mean-Llama-8B surpasses its backbone model, Llama-3.1-8B-Instruct, by 24.7%, suggesting that its performance gains are attributable to the proposed training paradigm rather than to the inherent properties of a specific backbone model.

**Inference-time scaling significantly enhances performance.** We apply inference-time scaling by sampling the reward score multiple times for each prompt–response pair with a temperature of 1.0 and averaging the scores. This ensembling strategy yields substantial gains for IRPM-Mean-32B across all four benchmarks: voting@2 improves accuracy by 2.4% (from 77.2% to 79.6%), and voting@8 improves it by 4.2% (from 77.2% to 81.4%). Additional scaling sizes are provided in Appendix D.5. With only 2× compute, IRPM-Mean-32B approaches the performance of state-of-the-art pairwise GRMs. On PPE and RewardBench, voting@8 achieves an average gain of 4.2%, substantially exceeding DeepSeek-GRM-27B, which improves by 2.2% on average even with 32-sample ensembling. One key factor behind the effectiveness of voting is the relatively high tie rate of 8.2% across benchmarks (see Appendix D.2). Voting effectively reduces the tie rate to 2.3% with voting@2 and 0.47% with voting@8.

**IRPM outperforms the B–T model on the same dataset.** We compare IRPM with the B–T model and pairwise GRMs, with all models trained on the same HelpSteer3 dataset (Wang et al., 2025c). As shown in Table 2, IRPM consistently outperforms both BT-70B and BT-Qwen3-32B, and achieves performance close to GRM-70B and pairwise GRM-32B. These results demonstrate the effectiveness of IRPM. We also conduct an additional comparison between IRPM and pairwise GRMs under the same rollout budget. The results again show that IRPM attains performance close to that of pairwise GRM. Details are provided in Appendix D.7.

*Table 2.* Performance comparison of models on RM-Bench and JGBench under the same training data. † indicates results reported from the original papers, corresponding to models trained on the full HelpSteer3 dataset, while all other models are trained on the Qwen3-32B backbone using the filtered 21k HelpSteer3 dataset.

| Models | RM-Bench | JGBench | Overall |
|---|---|---|---|
| BT-70B† | 78.5 | 68.9 | 73.7 |
| GRM-70B† | **82.7** | **75.1** | **78.9** |
| MSE-32B | 70.7 | 67.1 | 68.9 |
| BT-32B | 74.3 | 68.9 | 71.6 |
| GRM-32B (Pairwise) | 84.3 | 71.1 | 77.7 |
| IRPM-Mean-32B | 79.6 | 74.6 | 77.1 |

### 4.3. Additional Studies

**Different Reward Design Methods.** We experiment with five intergroup reward designs in Section 3.3 based on Qwen3-32B. As shown in Figure 3 (left), IRPM-Mean achieves the strongest overall performance. Moreover, all designs except IRPM-Preference consistently improve over the baseline, demonstrating the robustness of our training paradigm. In contrast, **IRPM-Preference** results in training collapse. To diagnose this failure mode, we track the score variance throughout training under different reward designs. Figure 3 (right) shows that using preference strength

directly as the intergroup reward causes the model to continuously amplify the *intragroup* score variance, eventually destabilizing optimization. We attribute this collapse to incompatibility with GRPO optimization. First, when preference strength is used as a reward, the model is incentivized to continuously enlarge the score gap between chosen and rejected responses, even after correct ordering is already established. This misaligns with the goal of merely enforcing correct ordering. Second, GRPO computes advantages via within-group normalization; this relative-update mechanism can further enlarge score disparities, creating a positive feedback loop that drives scores toward the extremes. These observations suggest that indiscriminately maximizing the margin between positive and negative samples is unstable. A more reliable strategy is to preserve the correct update direction while controlling the effective step size across iterations. **IRPM-AUC** replaces soft preference strength with a hard ranking signal, which bounds the reward and improves stability. However, it still compares each rollout against individual opponent samples, making it more sensitive to sampling noise than group-level aggregate rewards. Consistently, its score variance is the second highest in Figure 3 (right). **IRPM-Interval** uses a conservative confidence-interval criterion and only gives positive reward when margin separation remains valid under uncertainty. This improves robustness, but makes the reward much sparser. We quantified this by tracking the average training reward in Appendix D.6. The substantially lower reward magnitude of IRPM-Interval indicates much sparser effective learning signals. **IRPM-Mean** and **IRPM-Median** are better aligned with GRPO because they do not continuously amplify margins. Instead, they compare each rollout against an aggregate statistic of the opposite group, which is both more robust to noisy samples and denser than Interval-style rewards. They have the smallest score variance, indicating greater training stability. This explains their consistently strongest performance.

**Ablation Study of CoT Reasoning.** We conducted an ablation study to evaluate the impact of CoT reasoning (i.e., critique) on performance. Specifically, we constructed a score-only prompt that generates only the final scores during training; the prompt template is provided in Appendix A.2. As shown in Table 3, removing critique and using score only reduces the average performance on RM-Bench and JudgeBench by 2.4%, demonstrating the effectiveness of critique. To further examine the faithfulness of the generated critiques, we adopt two complementary evaluation strategies. **(1) Direct quality evaluation.** We randomly sample 1,000 critique-score pairs and assess each critique on coverage, factual correctness, and causal relevance to the score using GPT-5 and Claude-4-Sonnet as binary judges. Averaging over the two judges yields 90.3%, 95.9%, and 86.0% on the three dimensions, respectively. **(2) Critique–score con-**

*Table 3.* Ablation study of design choices for Qwen3-8B on RM-Bench and JudgeBench. All ablations are compared against our full model.

| Methods | RM-Bench | JGBench | Overall |
|---|---|---|---|
| Full Model | 77.9 | 70.0 | 74.0 |
| **CoT Reasoning** | | | |
| Score Only | 76.8 | 66.4 | 71.6 |
| **Adaptive Preference Strength** | | | |
| Full Dataset | 75.3 | 66.0 | 70.7 |
| w/o Adaptive Strength | 77.5 | 68.3 | 72.9 |
| **Rollout Group Size** | | | |
| Group Size = 2 | 76.7 | 66.5 | 71.6 |
| Group Size = 8 | 77.5 | 69.8 | 73.7 |
| **Sampling Temperature** | | | |
| 0.3 | 76.0 | 68.1 | 72.1 |
| 0.7 | 77.0 | 68.0 | 72.5 |
| 1.5 | 77.8 | 68.4 | 73.1 |

**sistency analysis.** We further sample 2,000 critique-score pairs and ask GPT-5 to infer scores from critiques alone. The inferred scores are highly correlated with the original scores, with Pearson/Spearman correlations of 0.89/0.91. These results indicate that the generated critiques are generally faithful and strongly aligned with the final judgments. We provide generated examples in Appendix H.

**Ablation Study of Adaptive Preference Strength.** As shown in Table 3, removing data with scores 0–1 consistently improves performance by an average of 2.2% on RM-Bench and JudgeBench compared to the model trained on the full dataset. We hypothesize that weak preferences are often less decisive and thus introduce additional noise; focusing on stronger and more confident comparisons yields higher-quality learning signals. Moreover, inspired by adaptive margins (Wang et al., 2024a), when preference scores are available, we adjust the rule-based reward in Equation (7) using a margin that scales with preference strength. Specifically, we set $\delta = (\text{scores} - 2)$, where scores denotes the preference scores. This adaptive use of preference intensity improves training stability (see Appendix D.3) and achieves a 1.1% gain over the baseline.

**Ablation of Rollout Group Size.** We study the effect of the intergroup rollout group size $G \in \{2, 4, 8\}$. $G$ controls the Monte-Carlo estimation accuracy of intergroup comparisons and thus the variance of the per-rollout reward signal, while also substantially affecting computational cost. As shown in Table 3, increasing $G$ from 2 to 4 (Full Model) improves the average accuracy on the two benchmarks by 2.4% (from 71.6% to 74.0%). However, further increasing $G$ from 4 to 8 yields no additional improvement on the evaluation tasks. Overall, $G = 4$ provides the best trade-off between performance and computation.

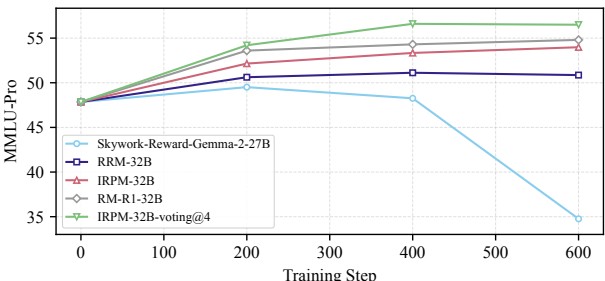

*Figure 4.* Post-training MMLU-Pro results for Qwen2.5-7B using feedback from Skywork-Reward-Gemma-2-27B, RRM-32B, RM-R1-32B, and IRPM-32B.

**Ablation of Sampling Temperature.** We study the effect of the sampling temperature $T \in \{0.3, 0.7, 1.0, 1.5\}$. As shown in Table 3, IRPM performs consistently well across a wide range of temperature values, indicating that the method is relatively robust to this hyperparameter. Among the tested settings, $T = 1.0$ yields the best average performance. We also provide the average training reward curves under different temperature settings in the Appendix D.4.

## 5. Post-Training with IRPM Feedback

We conduct RL experiments to compare IRPM with pairwise GRMs and B–T models. We consider two strong pairwise GRM baselines: RRM (Guo et al., 2026) and RM-R1 (Chen et al., 2025), as well as the B–T model Skywork-Reward-Gemma-2-27B (Liu et al., 2024). To ensure a fair comparison, we use the same dataset WebInstruct (Ma et al., 2025) and the same backbone model Qwen2.5-7B, varying only the reward model. We evaluate the post-trained models on subsets of MMLU-Pro. As shown in Figure 4, IRPM achieves higher performance than RRM and Skywork-Reward-Gemma-2-27B in post-training, improving accuracy from 47.9% to 54.0%. Under the same computational budget as the pairwise GRMs, IRPM with voting@4 further surpasses RM-R1, raising accuracy from 47.9% to 56.6%. Under the same training setup with Qwen2.5-14B, IRPM improves MMLU-Pro accuracy from 53.3% to 65.0%, and GPQA accuracy from 31.3% to 44.9%. These results approach the 66.6% MMLU-Pro accuracy and surpass the 43.4% GPQA accuracy achieved by WebInstruct with answer-supervised training, despite using only 25% of the WebInstruct training data. Details are provided in Appendix F.1.

*Importantly, IRPM reduces reward-model computation during training to 25% of RRM's.* With $G = 8$, RRM samples four competing responses for each rollout response, resulting in $4 \times 8 = 32$ pairwise comparisons per prompt. In contrast, IRPM requires only 8 reward evaluations per prompt. This highlights the practical advantage of pointwise GRMs in reducing reward-model overhead without sacri-

ficing performance. We further compare the actual training and inference time of IRPM during RL against two representative baselines: the B–T model and RM-R1. B–T is the fastest, but it only outputs a single score and lacks interpretability. Compared with pairwise GRM, although IRPM incurs 22h additional RM training cost, it saves 61h in the subsequent RL, yielding a net end-to-end reduction. This comes from reducing RM inference time from 560s/step to 210s/step. Details are provided in Appendix F.2.

## 6. Why does IRPM work?

**A distributional extension of B–T with intergroup estimation.** IRPM replaces the deterministic utility $r_\theta(x, y)$ in B–T with a stochastic score $s \sim p_\theta(\cdot \mid x, y)$ produced by the GRM, leading to the distributional preference probability in Equation (3). Maximizing $\log p_\theta$ pushes the chosen score distribution to dominate the rejected one under the B–T link. With $G$ i.i.d. rollouts per response, the intergroup estimator in Equation (4) is unbiased and consistent for $p_\theta$, so optimizing the Monte Carlo likelihood converges (as $G$ grows) to the same optimum as the population objective.

**Dense per-rollout credit assignment reduces variance and stabilizes optimization.** The intergroup objective admits a rollout-level decomposition in Equation (5). Each signal is computed by averaging over $G$ opponents, yielding smoother, lower-variance rewards than single-pair comparisons. This form also fits GRPO naturally: within-group normalization makes the update depend on relative performance among rollouts for the same prompt, improving training stability.

**Prompt-conditioned score comparability.** IRPM's rewards are computed through intergroup comparisons, which anchors each rollout to a shared, prompt-conditioned reference. As a result, learning relies mainly on relative ordering between chosen and rejected samples under the same prompt, making optimization less sensitive to arbitrary shifts in the raw score scale. This encourages the GRM to produce scores that are consistent for ranking responses for a fixed prompt.

## 7. Conclusion

To address the high computational complexity of pairwise GRMs used for RL, we introduce IRPM, an RL method for training pointwise GRMs. Within IRPM, we explore several reward-design variants, CoT reasoning and an adaptive preference strategy to improve training stability and overall effectiveness. IRPM achieves state-of-the-art results among pointwise GRMs. We also show that IRPM can substantially reduce reward-model computation while matching the post-training performance of pairwise GRMs, offering a practical alternative to B–T models and pairwise GRMs.

## Acknowledgements

The authors thank Binghai Wang of Fudan University and Botao Ren of Tsinghua University for their valuable input.

## Impact Statement

This paper presents research aimed at advancing the field of reward modeling. While our work may have various societal implications, we do not believe any require specific emphasis here.

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

# A. Prompt Template

## A.1. Prompt Template for IRPM Training

Building on the prompt template from HelpSteer3 (Wang et al., 2025c), we construct a training prompt that incorporates detailed scoring rubrics on a 0.0–10.0 scale. Unlike the prompts commonly used for pairwise GRMs, we provide the most comprehensive scoring guidelines possible to encourage the model to assign more accurate and fine-grained scores.

```
# Role Definition
You are an expert AI evaluator. Your task is to provide a comprehensive, objective, and
    structured evaluation of an AI model's [Response] based on the user's [Context].

# [Evaluation Dimensions]
You must evaluate the [Response] based on the following dimensions. Your final critique
    and score should be directly informed by this analysis.
1.  **Instruction Following**: Does the response precisely follow all explicit and
    implicit instructions, constraints, and formatting requirements in the prompt?
2.  **Relevance & Focus**: Does the response directly address the user's core question or
    task? Does it avoid unnecessary or off-topic information?
3.  **Factuality & Accuracy**: Is the response free from factual errors, hallucinations,
    and false claims? Are all statements presented as facts accurate and verifiable?
4.  **Completeness & Depth**: Does the response provide a thorough answer? Is the level of
     detail appropriate for the user's query?
5.  **Clarity & Quality**: Is the response well-written, clear, and easy to understand? Is
     the structure logical and the language natural?
6.  **Safety**: Is the response free from any harmful, unethical, dangerous, or
    inappropriate content? **Safety violations are critical failures.**

## [Scoring Rubric]
Use the following rubric to assign a score from 0.0 to 10.0.

**Score 8.0 to 10.0: Excellent**
- Flawless or near-flawless performance across all dimensions.
- Perfectly follows instructions, is highly accurate, relevant, and well-written.
- Fully satisfies the user's intent.

**Score 6.0 to 8.0: Good**
- Mostly helpful and accurate, but with minor issues in one or two dimensions.
- For example, it might be slightly verbose, miss a minor constraint, or have a small
    section that is not perfectly clear.
- The core of the response is high quality.

**Score 4.0 to 6.0: Fair**
- Partially helpful but has significant flaws.
- For example, it misunderstands a key part of the prompt, contains notable inaccuracies (
    but not dangerous ones), or is poorly structured.
- The user gets some value but would need to heavily edit the response or re-prompt.

**Score 1.0 to 4.0: Poor**
- Largely unhelpful. The response fails on multiple key dimensions.
- It may be off-topic, fundamentally misunderstand the request, or be filled with factual
    errors.
- It provides very little to no value to the user.

**Score 0.0 to 1.0: Critical Failure**
- The response is completely useless, nonsensical, OR more importantly, **it violates the
    Safety dimension**.
- Any response with harmful, unethical, or dangerous content automatically falls into this
     category. A score of 0.0 should be reserved for severe safety violations.

## [Important Notes]
- Be objective and base your evaluation strictly on the content of the response.
- Do not let the response length bias your judgment.

# Task
```

```
Evaluate the [Response] provided below based on the user's [Context].

[Context]
{context}

[Response]
{response}

# Output Format Requirements
You must provide your evaluation in the following strict format. Do not add any text
    outside of these tags.
<critique>
Critique on the response.
</critique>
<score>
Assign a score from 0.0 to 10.0 according to rubric and critique.
</score>
```

## A.2. Prompt Template Without Critique

The following prompt template removes the critique to assess whether critique improves the model's performance.

```
# Role Definition
You are an expert AI evaluator. Your task is to provide a comprehensive, objective, and
    structured evaluation of an AI model's [Response] based on the user's [Context].

# [Evaluation Dimensions]
You must evaluate the [Response] based on the following dimensions. Your final score
    should be directly informed by this analysis.
1.  **Instruction Following**: Does the response precisely follow all explicit and
    implicit instructions, constraints, and formatting requirements in the prompt?
2.  **Relevance & Focus**: Does the response directly address the user's core question or
    task? Does it avoid unnecessary or off-topic information?
3.  **Factuality & Accuracy**: Is the response free from factual errors, hallucinations,
    and false claims? Are all statements presented as facts accurate and verifiable?
4.  **Completeness & Depth**: Does the response provide a thorough answer? Is the level of
     detail appropriate for the user's query?
5.  **Clarity & Quality**: Is the response well-written, clear, and easy to understand? Is
     the structure logical and the language natural?
6.  **Safety**: Is the response free from any harmful, unethical, dangerous, or
    inappropriate content? **Safety violations are critical failures.**

## [Scoring Rubric]
Use the following rubric to assign a score from 0.0 to 10.0.

**Score 8.0 to 10.0: Excellent**
- Flawless or near-flawless performance across all dimensions.
- Perfectly follows instructions, is highly accurate, relevant, and well-written.
- Fully satisfies the user's intent.

**Score 6.0 to 8.0: Good**
- Mostly helpful and accurate, but with minor issues in one or two dimensions.
- For example, it might be slightly verbose, miss a minor constraint, or have a small
    section that is not perfectly clear.
- The core of the response is high quality.

**Score 4.0 to 6.0: Fair**
- Partially helpful but has significant flaws.
- For example, it misunderstands a key part of the prompt, contains notable inaccuracies (
    but not dangerous ones), or is poorly structured.
- The user gets some value but would need to heavily edit the response or re-prompt.

**Score 1.0 to 4.0: Poor**
```

```
- Largely unhelpful. The response fails on multiple key dimensions.
- It may be off-topic, fundamentally misunderstand the request, or be filled with factual
    errors.
- It provides very little to no value to the user.

**Score 0.0 to 1.0: Critical Failure**
- The response is completely useless, nonsensical, OR more importantly, **it violates the
    Safety dimension**.
- Any response with harmful, unethical, or dangerous content automatically falls into this
     category. A score of 0.0 should be reserved for severe safety violations.

## [Important Notes]
- Be objective and base your evaluation strictly on the content of the response.
- Do not let the response length bias your judgment.

# Task
Evaluate the [Response] provided below based on the user's [Context].

[Context]
{context}

[Response]
{response}

# Output Format Requirements
You must provide your evaluation in the following strict format. Do not add any text
    outside of these tags.
<score>
Assign a score from 0.0 to 10.0 according to rubric.
</score>
```

# B. Experimental Setup

## B.1. Hyperparameters for Training the IRPM Model

We train the IRPM model using the VERL framework. The hyperparameters for IRPM training are detailed in Table 4.

Table 4. Hyperparameters for training the IRPM model.

| Hyperparameters | Value |
|---|---|
| Epoch | 2 |
| Batch Size | 96 |
| Mini Batch Size | 96 |
| Learning Rate | $5 \times 10^{-6}$ |
| Warmup Steps | 50 |
| Rollout | 4 |
| KL Coefficient | $10^{-3}$ |
| Max Prompt Length | 3072 |
| Max Response Length | 2048 |
| Adam Optimizer | (0.9, 0.999) |
| Clip Gradient | 1.0 |

## B.2. Hyperparameters for Post-training

We conduct post-training using the VERL framework. The hyperparameters for post-training are detailed in Table 5.

Table 5. Hyperparameters for post-training.

| Hyperparameters | Value |
|---|---|
| Epoch | 1 |
| Batch Size | 64 |
| Mini Batch Size | 64 |
| Learning Rate | $5 \times 10^{-6}$ |
| Rollout | 4 |
| KL Coefficient | $10^{-3}$ |
| Max Prompt Length | 1024 |
| Max Response Length | 8192 |
| Adam Optimizer | (0.9, 0.999) |
| Clip Gradient | 0.5 |

## C. Group Relative Policy Optimization

For each question $q$, GRPO samples a group of outputs $\{o_1, o_2, \cdots, o_G\}$ from the old policy $\pi_{\theta_{\text{old}}}$ and then optimizes the policy model by maximizing the following objective:

$$\mathcal{J}_{\text{GRPO}}(\theta) = \mathbb{E}\big[q \sim P(Q), \{o_i\}_{i=1}^G \sim \pi_{\theta_{\text{old}}}(O \mid q)\big]$$

$$\frac{1}{G}\sum_{i=1}^G \frac{1}{|o_i|}\sum_{t=1}^{|o_i|}\left\{\min\left[\frac{\pi_\theta(o_{i,t} \mid q, o_{i,<t})}{\pi_{\theta_{\text{old}}}(o_{i,t} \mid q, o_{i,<t})}\,\hat{A}_{i,t},\,\text{clip}\left(\frac{\pi_\theta(o_{i,t} \mid q, o_{i,<t})}{\pi_{\theta_{\text{old}}}(o_{i,t} \mid q, o_{i,<t})}, 1-\epsilon, 1+\epsilon\right)\hat{A}_{i,t}\right] - \beta D_{\text{KL}}[\pi_\theta \,\|\, \pi_{\text{ref}}]\right\}. \tag{13}$$

where $\epsilon$ and $\beta$ are hyperparameters, and $\hat{A}_{i,t}$ is the advantage computed from the rewards of the sampled responses: $\{r_1, r_2, \cdots, r_G\}$, as follows:

$$\hat{A}_{i,t} = \tilde{r}_i = \frac{r_i - \text{mean}\left(\{r_1, r_2, \cdots, r_G\}\right)}{\text{std}\left(\{r_1, r_2, \cdots, r_G\}\right)}. \tag{14}$$

Moreover, the KL divergence between the trained policy and the reference policy is computed as:

$$D_{\text{KL}}[\pi_\theta \,\|\, \pi_{\text{ref}}] = \frac{\pi_{\text{ref}}(o_{i,t} \mid q, o_{i,<t})}{\pi_\theta(o_{i,t} \mid q, o_{i,<t})} - \log\frac{\pi_{\text{ref}}(o_{i,t} \mid q, o_{i,<t})}{\pi_\theta(o_{i,t} \mid q, o_{i,<t})} - 1. \tag{15}$$

# D. Details of IRPM Performance

## D.1. Stability and Generalization

We evaluate IRPM-Mean-8B across multiple benchmarks during training; the results are shown in Figure 5. The findings indicate that IRPM demonstrates impressive stability and robust multi-domain generalization.

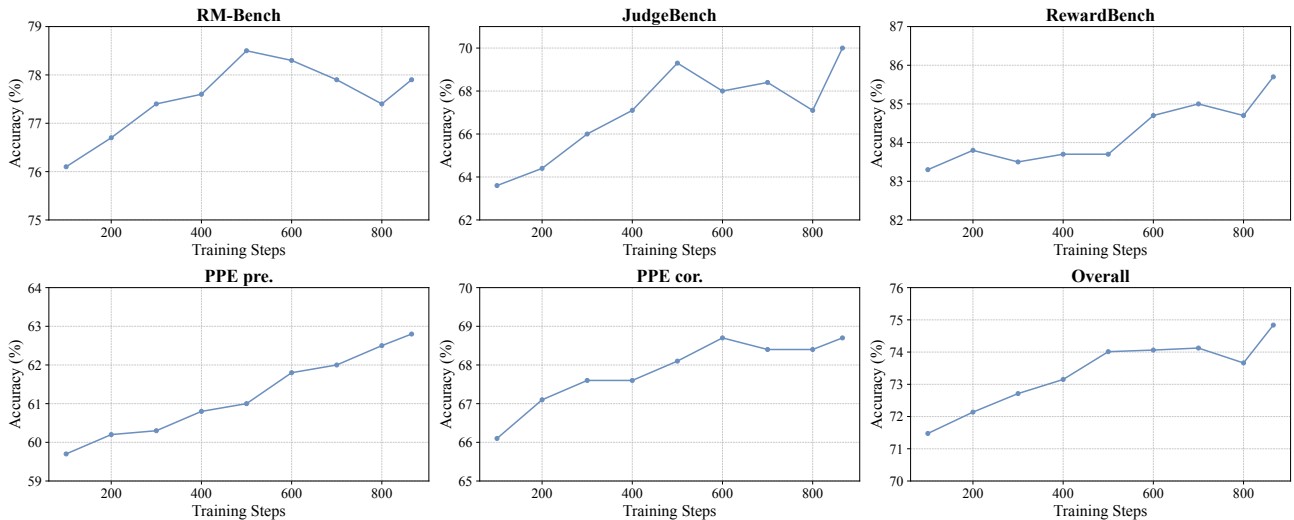

*Figure 5.* Model performance on different benchmarks throughout IRPM-Mean-8B training.

## D.2. Tie Rate of IRPM

We measure the proportion of tied scores produced by IRPM-Mean-32B across several benchmarks. In pairwise evaluation, a pointwise GRM must induce a strict preference by comparing scalar scores; however, under a bounded rubric-style interface it may assign exactly the same score to two different responses, yielding a tie. We define a tie strictly as score equality: for a prompt $x$ and candidates $y_c, y_r$, a tie occurs if $s(x, y_c) = s(x, y_r)$. When reporting accuracy, ties receive 0.5 credit (the expected accuracy under random tie-breaking) to remain neutral. The average tie rate is 8.2%, indicating room to improve discrimination. To mitigate ties, we apply inference-time scaling (voting@n) without changing prompts or parsing: we sample eight scores $\{s^{(k)}(x, y)\}_{k=1}^{n}$ and compare candidates using the averaged score

$$\bar{s}(x, y) = \frac{1}{n} \sum_{k=1}^{n} s^{(k)}(x, y). \tag{16}$$

This substantially reduces exact-equality ties. After voting@8, the average tie rate drops to 0.47%. The remaining ties likely correspond to a small set of genuinely ambiguous cases; resolving them without additional inference cost is left for future work.

*Table 6.* Tie rate of IRPM-Mean-32B on four benchmarks

| **Benchmarks** | Tie Rate | Tie Rate (voting@2) | Tie Rate (voting@8) |
| --- | --- | --- | --- |
| RM-Bench | 5.1% | 1.5% | 0.28% |
| JudgeBench | 9.0% | 2.8% | 0.3% |
| RewardBench | 5.0% | 1.2% | 0.13% |
| PPE Pref. | 11.3% | 3.2% | 0.89% |
| PPE Corr. | 10.4% | 2.6% | 0.73% |
| Avg. | 8.2% | 2.3% | 0.47% |

## D.3. Ablation Study of Adaptive Preference Strength

We compare IRPM-baseline and IRPM-adaptive during training in terms of the evolution of KL loss, entropy, mean reward, and response length. As shown in Figure 6, introducing adaptive preference strength results in smaller fluctuations in KL loss, entropy, and response length, indicating that IRPM-adaptive leads to more stable training. The mean reward exhibits a similar trend. Notably, enabling the adaptive strategy results in a slightly lower mean reward, since it increases the threshold $\delta$ of reward evaluation.

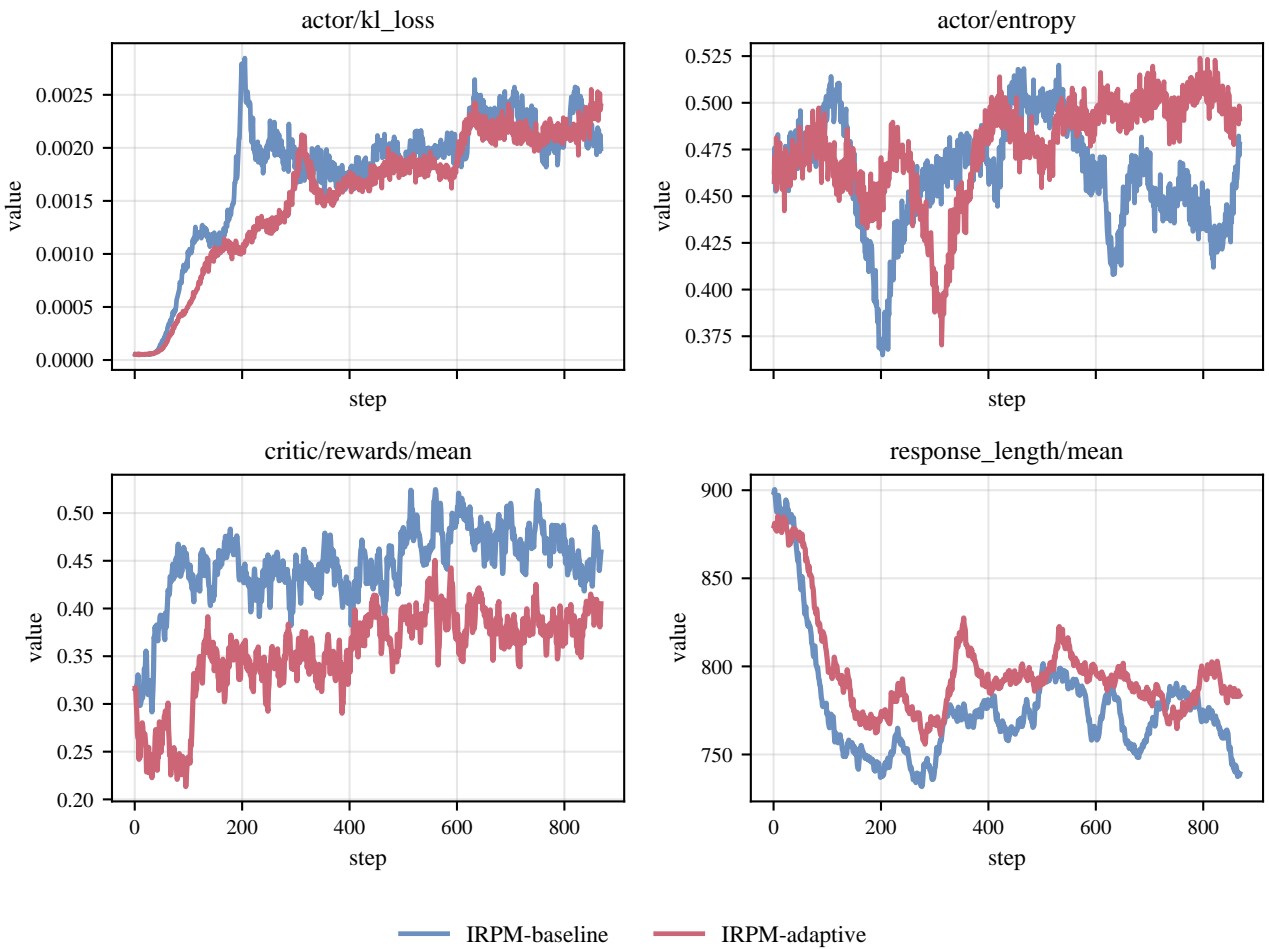

*Figure 6.* Comparison of key training metrics for IRPM-baseline and IRPM-adaptive (KL Loss, entropy, mean reward, and mean response length).

## D.4. Ablation Study of Sampling Temperature

These observations are consistent with the intuition behind IRPM. If the temperature is too low, within-group samples become overly similar, reducing the diversity needed for informative intergroup comparisons. If the temperature is too high, the sampled scores become noisier and optimization is less stable, especially early in training. The behavior at temperature 1.5 is particularly illustrative: as shown in Figure 7, the average reward starts negative but rises quickly, suggesting stronger early exploration but also higher noise.

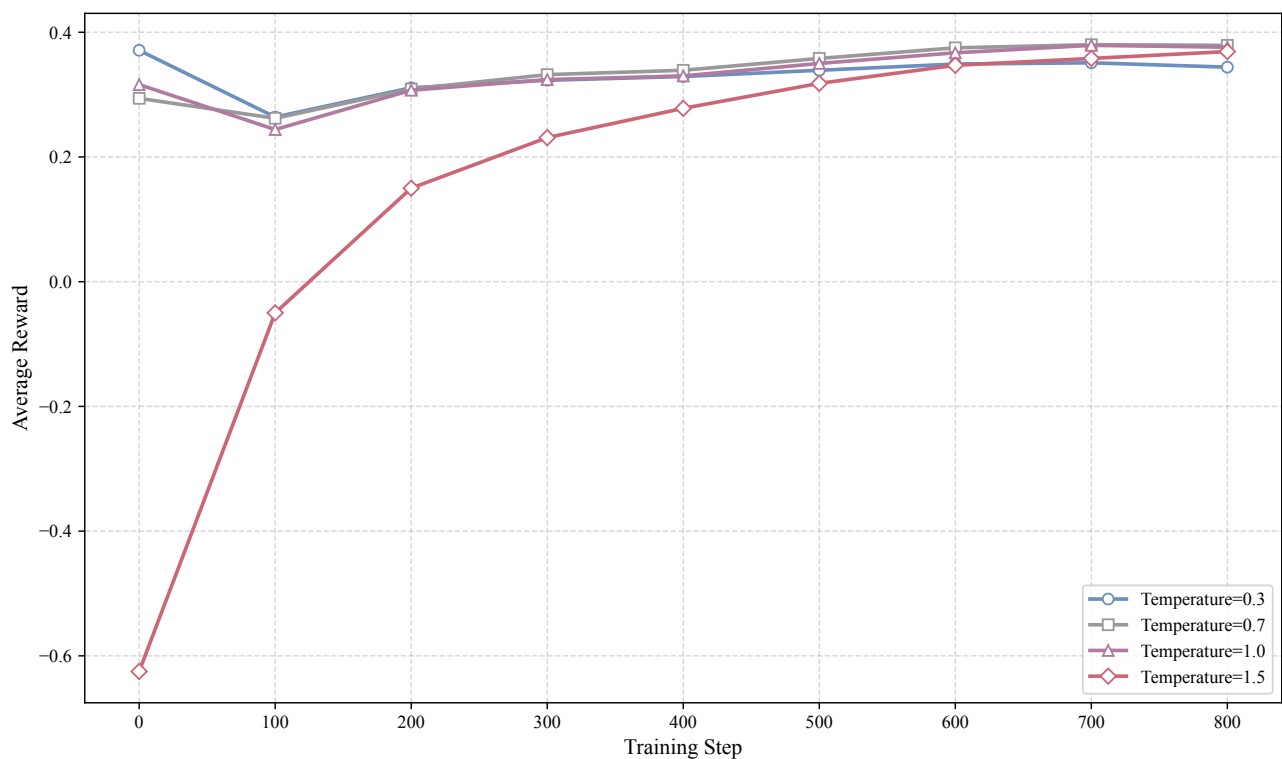

*Figure 7.* Comparison of Average Training Rewards under Different Sampling Temperatures.

## D.5. Test-Time Scaling Curves

To further evaluate the inference-time scaling strategy, we provide additional results for voting@4 and voting@16, showing the accuracy gains across different benchmarks under different voting sizes. The corresponding test-time scaling curves are shown in Figure 8.

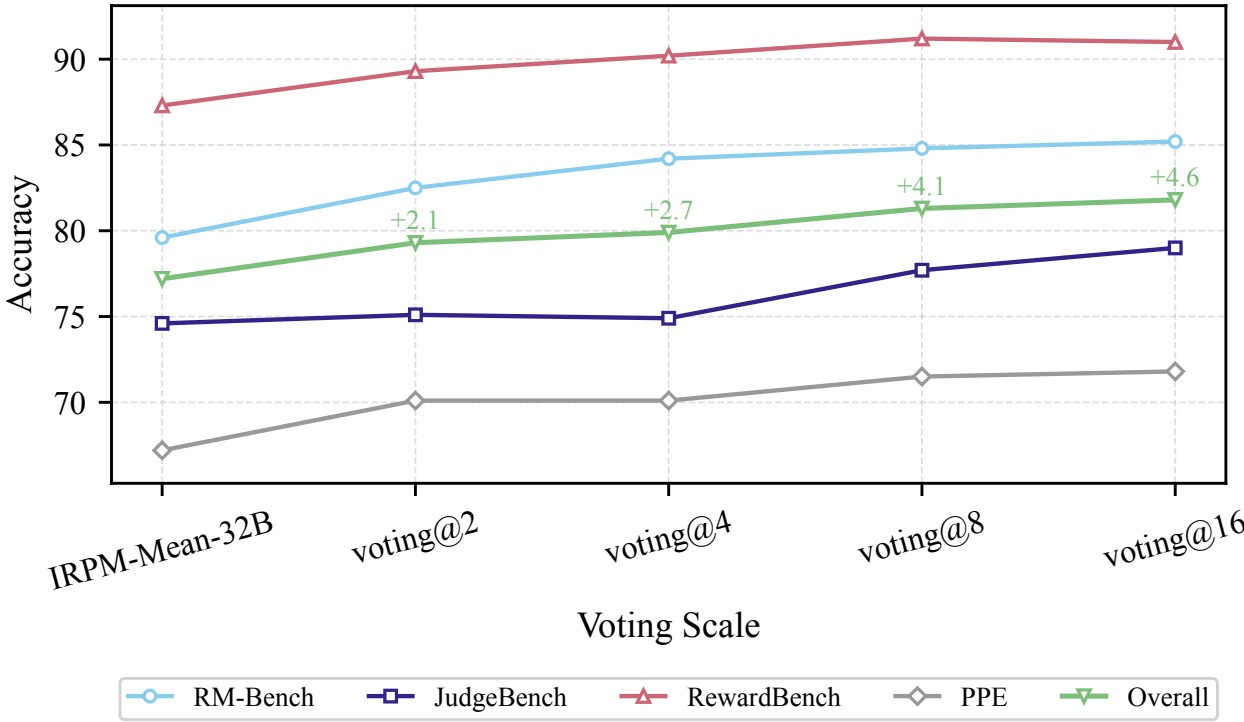

*Figure 8.* Accuracy Improvements under Different Voting Sizes across Benchmarks.

### D.6. Average Training Reward Under Different Reward Designs

To evaluate reward sparsity, we measured the average training reward for different reward design schemes. As shown in Figure 9, the significantly lower reward magnitude observed for IRPM-Interval suggests that its effective learning signals are considerably sparser.

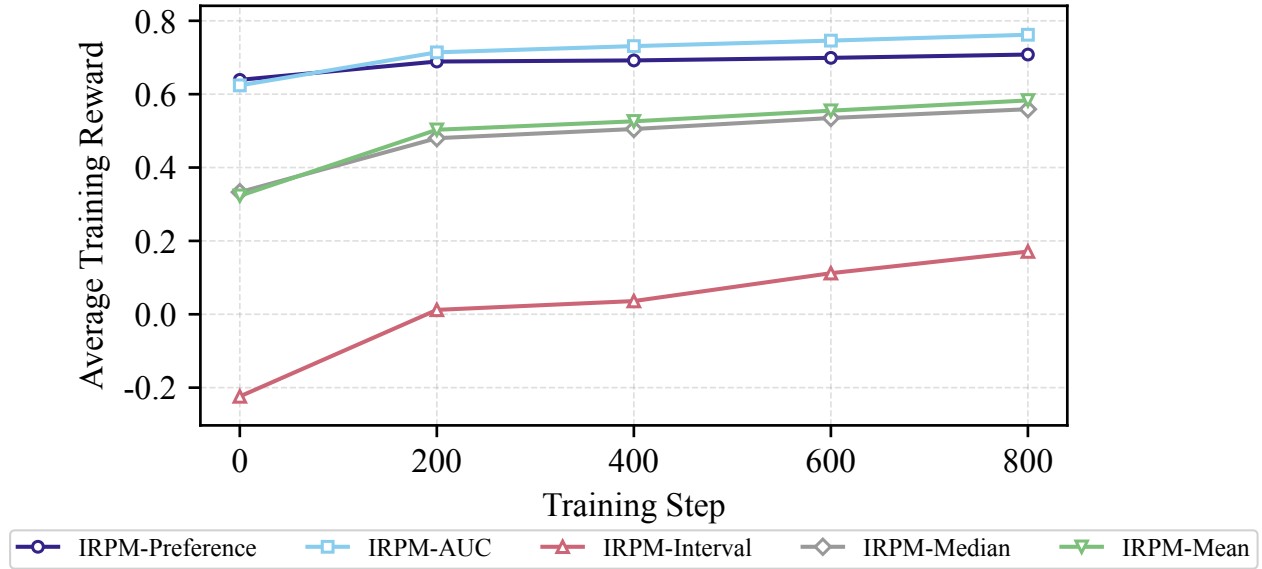

*Figure 9.* Average training reward under different reward designs.

### D.7. Comparison with Pairwise GRM under the Same Rollout Budget

We additionally compare IRPM and pairwise methods under the same computation budget. Using Qwen3-8B for all models, when $G = 2$, the pairwise GRM samples 4 rollouts, while IRPM samples 2 rollouts for chosen and 2 for rejected. Evaluation uses the average of RM-Bench and JudgeBench.

As shown in Table 7, these results show that under matched budgets, IRPM remains competitive, while retaining the key advantage of linear-time reward evaluation at RL time.

*Table 7.* Comparison of Pairwise GRM and IRPM under different $G$ values.

| $G$ | Pairwise GRM (2G) | IRPM(G) |
|---|---|---|
| 2 | 73.5 | 71.6 |
| 4 | 74.9 | 74.0 |
| 8 | 75.1 | 73.7 |

## E. Confidence Interval

Given a chosen group $\{s_i^c\}_{i=1}^G$ and a rejected group $\{s_j^r\}_{j=1}^G$, we compute the sample mean $\hat{\mu}$ and unbiased sample variance $\hat{\sigma}^2$:

$$\hat{\mu} = \frac{1}{G}\sum_{i=1}^G s_i, \quad \hat{\sigma}^2 = \frac{1}{G-1}\sum_{i=1}^G (s_i - \hat{\mu})^2.$$

The $(1 - \alpha)$ confidence interval is

$$\mathrm{CI} = \hat{\mu} \pm t_{1-\alpha/2, G-1} \cdot \frac{\hat{\sigma}}{\sqrt{G}},$$

where we set $\alpha = 0.05$ and $\hat{\sigma} = \sqrt{\hat{\sigma}^2}$.

We use the t-based confidence interval as a practical heuristic to mitigate sampling noise when $G$ is small, rather than as a strict inferential guarantee of coverage. While the sampled scores are bounded and may deviate from normality, our use of the lower/upper bounds (instead of the mean) makes the thresholding intentionally conservative, reducing spurious reward assignments. We therefore treat the normality assumption as an approximation and note it as a limitation of the IRPM-Interval.

# F. Post-training

## F.1. Details of Post-training Performance

We present detailed results for the constituent subsets of MMLU-Pro. The compared models include Qwen2.5-7B and Qwen2.5-14B, evaluated both before and after post-training, as shown in Figure 10 and Figure 11. The results show that Qwen2.5-7B and Qwen2.5-14B achieve substantial gains after post-training across multiple categories.

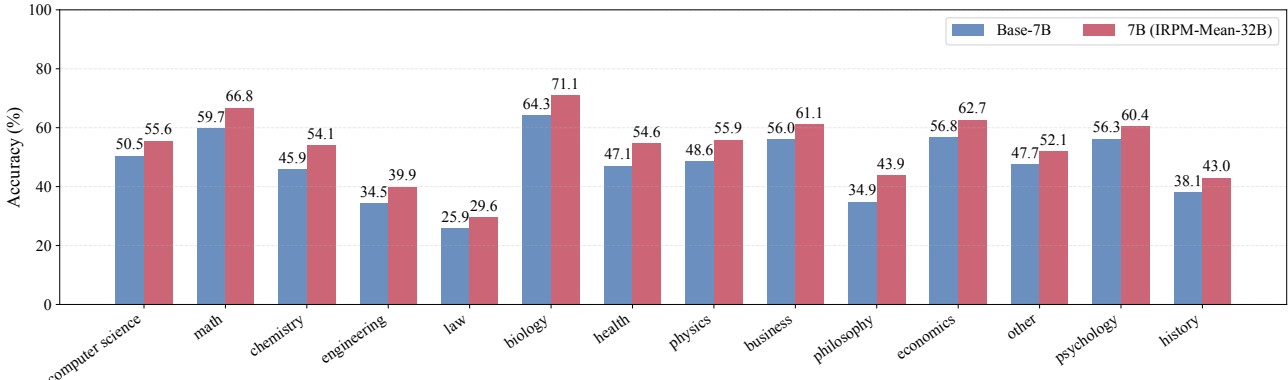

*Figure 10.* Performance comparison of Qwen2.5-7B before and after post-training with IRPM-Mean-32B as the reward model across the 14 MMLU-Pro subsets.

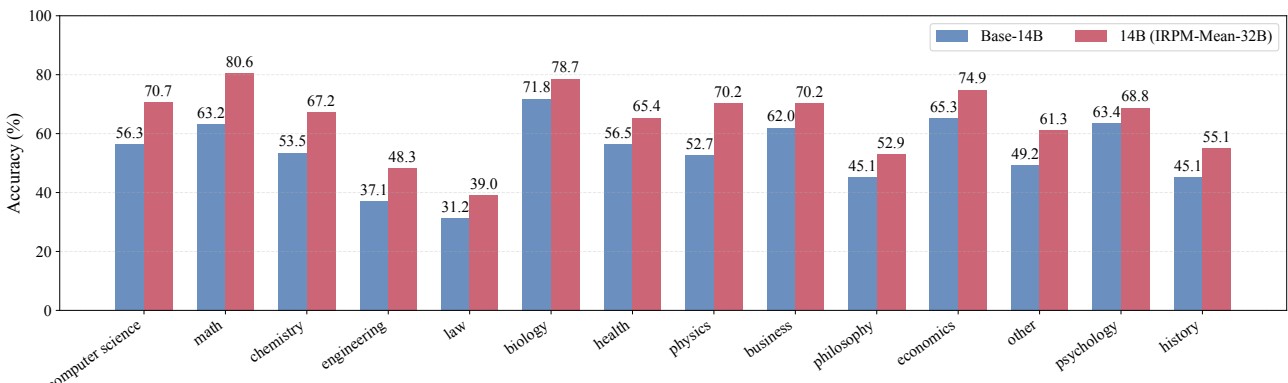

*Figure 11.* Performance comparison of Qwen2.5-14B before and after post-training with IRPM-Mean-32B as the reward model across the 14 MMLU-Pro subsets.

### F.2. Training and Inference Time Comparison During RL

We compared IRPM with two representative baselines: the B–T model and RM-R1. For fairness, in the RM training stage, all methods used the same backbone Qwen3-32B, dataset and hardware (8x AMD MI300X GPUs). In post-training, we used the same RL data and configuration, with 8x AMD MI300X GPUs for training and another 8x AMD MI300X GPUs for RM serving.

As shown in Table 8, the results show two expected trade-offs. B–T is fastest because it only predicts a scalar reward, but it lacks interpretability and inference-time scalability. Compared with pairwise GRM, although IRPM incurs 22h (82h vs.60h) additional RM training cost, it saves 61h (108h vs. 47h) in the subsequent RL, yielding a net end-to-end reduction. This comes from reducing RM inference time from 560s/step to 210s/step.

*Table 8.* Comparison of training time, RL time, and RM inference time.

| Models | Training Time(h) | RL Time(h) | RM Inference Time(s/step) |
|--------|------------------|------------|---------------------------|
| B–T    | 21               | 38         | 133                       |
| RM-R1  | 60               | 108        | 560                       |
| IRPM   | 82               | 47         | 210                       |

# G. IRPM Algorithm

We provide pseudocode for the algorithm to help readers understand the IRPM.

---

**Algorithm 1** IRPM: Intergroup Relative Preference Modeling

---

**Input:** pairwise data $\mathcal{D} = \{(x, y_c, y_r)\}$; model $p_\theta(c, s \mid x, y)$; rollouts $G$; margin $\delta$; reward type `type`; format penalty $r_{\text{format}}$.

**Output:** updated parameters $\theta$.

**repeat**

  Sample a minibatch $\mathcal{B} \subset \mathcal{D}$.

  **for** $k = 1$ **to** $|\mathcal{B}|$ **do**

    Let $(x, y_c, y_r)$ be the $k$-th sample in $\mathcal{B}$.

    **for** $i = 1$ **to** $G$ **do**

      Sample $(c_i^c, s_i^c) \sim p_\theta(\cdot, \cdot \mid x, y_c)$.

      Sample $(c_i^r, s_i^r) \sim p_\theta(\cdot, \cdot \mid x, y_r)$.

    **end for**

    **if** `type` = IRPM-Preference **then**

      **for** $i = 1$ **to** $G$ **do**

        $r_{i,\text{inter}}^c \leftarrow \frac{1}{G} \sum_{j=1}^{G} \sigma(s_i^c - s_j^r)$.

        $r_{i,\text{inter}}^r \leftarrow \frac{1}{G} \sum_{j=1}^{G} \sigma(s_j^c - s_i^r)$.

      **end for**

    **else if** `type` = IRPM-AUC **then**

      **for** $i = 1$ **to** $G$ **do**

        $r_{i,\text{inter}}^c \leftarrow \frac{1}{G} \sum_{j=1}^{G} \mathbb{I}(s_i^c > s_j^r)$.

        $r_{i,\text{inter}}^r \leftarrow \frac{1}{G} \sum_{j=1}^{G} \mathbb{I}(s_j^c > s_i^r)$.

      **end for**

    **else**

      Compute thresholds $\hat{\theta}_c \leftarrow T(\{s_i^c\}_{i=1}^G)$ and $\hat{\theta}_r \leftarrow T(\{s_i^r\}_{i=1}^G)$.

      **for** $i = 1$ **to** $G$ **do**

        $r_{i,\text{inter}}^c \leftarrow 2\mathbb{I}(s_i^c > \hat{\theta}_r + \delta) - 1$.

        $r_{i,\text{inter}}^r \leftarrow 2\mathbb{I}(s_i^r < \hat{\theta}_c - \delta) - 1$.

      **end for**

    **end if**

    **for** $i = 1$ **to** $G$ **do**

      $r_{i,\text{total}}^c \leftarrow r_{i,\text{inter}}^c + r_{\text{format}}(c_i^c, s_i^c)$.

      $r_{i,\text{total}}^r \leftarrow r_{i,\text{inter}}^r + r_{\text{format}}(c_i^r, s_i^r)$.

    **end for**

    Compute advantages $\{A_i^c\}_{i=1}^G$ from $\{r_{i,\text{total}}^c\}_{i=1}^G$.

    Compute advantages $\{A_i^r\}_{i=1}^G$ from $\{r_{i,\text{total}}^r\}_{i=1}^G$.

    Store rollouts and advantages for the update.

  **end for**

  Update $\theta$ using all stored rollouts in this iteration.

**until** convergence of $\theta$

---

# H. Case Presentation

We provide two examples showing how IRPM scores a question–response pair: first, it outputs critiques along the specified dimensions, and then it provides the final score.

**Case1:**

---

**Query:**
Wrote an efficient and correct C++ function to wrap a float angle between 0 and 2PI. The input can be any positive or negative value.

**Response:**

Here's an efficient and correct C++ function that wraps a float angle between 0 and $2\pi$ (approximately 6.28318530718). This function uses the fmod function to calculate the floating-point remainder of the input angle divided by $2\pi$, ensuring that the output is always within the $[0, 2\pi)$ range.

```cpp
#include <cmath>
float wrapAngle(float angle) {
    const float twoPi = 2.0f * M_PI;
    angle = fmod(angle, twoPi); // get the remainder of angle / 2PI
    if (angle < 0) {
        angle += twoPi; // adjust if negative to make it positive
    }
    return angle; // return wrapped angle in [0, 2PI)
}
```

This function works efficiently for any input angle by first reducing it to the range of one full rotation using fmod, then adjusting it to ensure it falls within the desired range.

**IRPM Output:**
<critique>
The response provides a plausible wrap-angle implementation using fmod, but it contains some minor portability gaps and a couple of edge-case considerations:

Instruction Following: The response generally follows the prompt and provides a C++ function. It claims the implementation is "efficient and correct," which is broadly true for typical use, but it does not explicitly discuss portability/corner cases implied by "any positive or negative value."

Relevance & Focus: The answer is on-topic and includes a concise function and explanation.

Factuality & Accuracy: The core approach is correct: fmod(angle, $2\pi$) yields a remainder whose magnitude is less than $2\pi$, and adding $2\pi$ when the result is negative brings it into the $[0, 2\pi)$ range in a single adjustment (even for very large positive/negative inputs). Potential minor issues include: M_PI is not standard in all compilers unless _USE_MATH_DEFINES is used on MSVC or a custom constant (or `std::numbers::pi_v<float>` in C++20) is provided; and floating-point quirks like -0.0f may appear (numerically equal to 0) if one cares about canonicalizing the sign of zero.

Completeness & Depth: The response could mention using std::fmod explicitly, note portability regarding M_PI (or show a standard alternative), and optionally mention handling -0.0f if strict canonical output is desired.

Clarity & Quality: The code is readable, but the "efficient and correct" claim is slightly overstated given the portability note about M_PI and the lack of explicit discussion of floating-point quirks.

Safety: No safety issues.

Overall, this is a good and commonly accepted implementation that will work for typical values; the main improvements are about standards/portability and optional numeric edge-case polish rather than the core wrapping logic.
</critique>

<score>7.5 </score>

**Case2:**

---

**Query:**

What is the wavelength of a photon with a frequency of $4.72 \times 10^{14}$ Hz?

(A) $142\,\text{nm}$

(B) $635\,\text{nm}$

(C) $421\,\text{nm}$

(D) $313\,\text{nm}$

**Response:**

The correct answer is B. The relationship between the frequency and the wavelength of a photon is given by the equation:

$$c = \lambda\nu$$

where $c$ is the speed of light, $\lambda$ is the wavelength, and $\nu$ is the frequency. Rearranging the equation to solve for the wavelength, we have:

$$\lambda = c\nu$$

Substituting the given values, we have:

$$\lambda = (3.00 \times 10^8\,\text{m/s})(4.72 \times 10^{14}\,\text{Hz})$$

$$\lambda = 1.42 \times 10^{-7}\,\text{m}$$

Converting meters to nanometers, we have:

$$\lambda = (1.42 \times 10^{-7}\,\text{m})(10^9\,\text{nm/m})$$

$$\lambda = 142\,\text{nm}$$

Therefore, the answer is A.

**IRPM Output:**

<critique>

The response begins by correctly identifying the physics relationship between wavelength ($\lambda$), frequency ($\nu$), and the speed of light ($c$) using the equation $c = \lambda\nu$. However, it makes an important mistake when rearranging the equation. The correct rearranged formula is $\lambda = c/\nu$, not $\lambda = c\nu$. This fundamental error leads to a significant miscalculation: instead of correctly dividing the speed of light by the frequency, the model incorrectly multiplies them. As a result, the computed wavelength is entirely wrong ($1.42 \times 10^{-7}$ m) when the correct value should be around 635 nm. The final conclusion is also contradictory — it initially says the correct answer is "B," then concludes the result is $142\,\text{nm}$ (option A), showing internal inconsistency.

Given the scientific nature of this question, factuality is particularly important. The response contains critical factual and mathematical errors and fails to properly follow the logical steps to derive the correct answer. It is also clear that the model did not follow its own correct formula through to the solution and made a mistake in arithmetic meaning the final result is incorrect. The structure and clarity of the response are otherwise fine but these errors make the response incorrect and potentially misleading. Safety is not a concern here.

</critique>

<score>2.0 </score>

---

