# OpenReview forum: "IRPM: Intergroup Relative Preference Modeling for Pointwise Generative Reward Models"
_ICML.cc/2026/Conference — ICML 2026 regular_

### Official Review · Reviewer_uugc · 2026-02-16

**Soundness:** 3
**Presentation:** 3
**Significance:** 2
**Originality:** 2
**Overall Recommendation:** 4
**Confidence:** 3

**Summary:**

This paper proposes IRPM (Intergroup Relative Preference Modeling), an RL-based method to train a pointwise generative reward model (GRM) from pairwise preference data. The key idea is the reformulation of Bradley–Terry model when utilities are stochastic scores sampled from a GRM, then using comparisons between a group of rollouts for the chosen response and a group for the rejected response to produce rollout-level rewards, optimized with GRPO-style updates. The claimed benefits are (i) pointless scores suitable for RLHF, (ii) reduced reward-evaluation complexity from O(n^2) of pairwise GRMs to O(n) and (iii) interpretability and inference-time scaling via generated critiques plus scores. Experiments show strong benchmark results for pointwise GRMs and competitive performance vs leading pairwise and pointwise GRMs as well as conventional BT reward models.

**Compliance With Llm Reviewing Policy:**

Affirmed.

**Final Justification:**

The authors response has addressed my questions and concerns.
I remain at my original rating which is in consensus with the other reviewers.

**Key Questions For Authors:**

See weaknesses

**Limitations:**

yes

**Strengths And Weaknesses:**

### Strengths
- Practical relevance. The O(n^2) inference complexity of pairwise GRMs is a real bottleneck which this paper addresses.
- The decomposition into rollout-level rewards and the intergroup Monte Carlo estimator are neat design choices.
- Empirical results are solid including evaluation against a wide range of baselines and several benchmark datasets. It also includes post-training experiments.
- The paper honestly reports the failure model of the IRPM-Preference method.

### Weaknesses
- Faithfulness of critiques is unvalidated. The paper claims interpretability due to the generated critiques. However, the analysis does not evaluate whether the critique content is truthful/causal vs merely a useful rationalization that enables generating a better score. The paper shows critiques help performance (score-only hurts, but the difference is only 1.8%). This is, however, not the same as trustworthy explanations.
- The most direct Monte Carlo B–T decomposition leads to training collapse. This leads to a question of how tightly the proposed variants of IRPM-Mean /-Median /-Interval /-AUC are connected with the original objective. Is there a lower-bound on optimality that could be derived?
- Although the paper emphasises O(n) complexity, the actual RL pipeline still requires: G rollouts, intergroup comparisons, critique + score generation. The paper does not provide wall-clock training and inference time comparisons against the alternative methods (the claims of 25% speed up from Section 5 refer to a very specific setup and focuses purely on the reward model computation during RL training, a more comprehensive analysis would be beneficial).

---

> ### Author Rebuttal · Authors · 2026-03-31
>
> Dear Reviewer uugc,
>
> Thank you for recognizing the computational efficiency of IRPM. Below are our responses to the weaknesses and questions you raised.
> ## W1: Faithfulness of Critiques
> We present the prompt used to generate critiques in Appendix A.1 and the generate critique content in Appendix H of the paper, so that readers can better understand the composition of the critiques. We conduct the following analyses to demonstrate critique faithfulness:
>
> **(1) Direct quantitative evaluation of critique quality.** We randomly sample 1k critique–score pairs from the benchmark and evaluate each critique along three dimensions:
> - Coverage: whether the critique covers the key aspects affecting the judgment;
> - Factual correctness: whether the stated pros/cons are supported by the candidate response rather than hallucinated;
> - Causal relevance to score: whether the critique provides reasons that can explain the final score.
>
> Using strong judge models (GPT-5 and Claude-4-Sonnet) to assign binary judgments (0/1), we average the scores from the two models:
>
> |Coverage(%) | Factual correctness(%) | Causal relevance to score(%) |
> |-|-|-|
> |90.3|95.9|86.0|
>
> These results provide partial evidence that IRPM critiques are generally faithful.
>
> **(2) Critique–score consistency analysis.** We additionally sample 2k critique–score pairs, ask GPT-5 to infer a score from the generated **critique alone**, and measure correlation with the model’s original score. The correlations are high:
> |Pearson | Spearman |
> |-|-|
> |0.893|0.909|
>
> This suggests that the critiques are strongly aligned with the produced scores.
>
> In the GRM literature, the interpretability and verifiability of critiques constitute an important field-level challenge. Still, we believe these analyses provide supportive evidence that the critiques are often informative and aligned with the score, although fully establishing causal faithfulness remains an open challenge.
> ## W2: Theoretical connection and optimality of IRPM variants
> The connection between the IRPM variants and the original stochastic B–T formulation is **principled**, with the main difference being a trade-off between **objective faithfulness** and **optimization stability**.
>
> First, Eq.(4) is the unbiased finite-sample Monte Carlo estimator of the stochastic B–T preference probability in Eq.(3), and Eq.(5) is an exact rollout-level decomposition of the same estimator. Therefore, **IRPM–Preference** is the most faithful finite-sample surrogate to the stochastic B–T comparison probability. Its collapse is thus not due to objective mismatch, but due to **optimization instability when this reward is combined with GRPO-style updates**, as discussed in Sec4.3.
>
> The remaining variants are stabilized surrogates derived from the same Monte Carlo intergroup comparison principle. **IRPM–AUC** keeps the full $G \times G$ comparison structure and only replaces the soft B–T kernel $\sigma(\Delta)$ with the hard ranking kernel $\mathbf{1}(\Delta > 0)$. **IRPM–Mean/Median/Interval** make a further approximation by replacing the full empirical opponent group with a representative statistic (mean, median, or confidence bound) plus a margin $\delta$. Hence, these variants are best viewed as **progressively more regularized, lower-variance approximations** to the same intergroup objective, rather than unrelated heuristics.
>
> One can derive a generic surrogate-gap bound of the following form. Let $J(\theta)$ denote the expected stochastic B–T objective and $J_v(\theta)$ the surrogate induced by variant $v$. If  $\theta^* = \arg\max_\theta J(\theta)$ and $\theta_v^* = \arg\max_\theta J_v(\theta)$, then
> $$
> J(\theta_v^\*) \geq J(\theta^\*) - 2 \sup_{\theta} |J(\theta) - J_v(\theta)|.
> $$
> Therefore, the key issue is how tightly each variant approximates the original objective.
>
> For **IRPM–AUC**, this connection is especially clean because only the comparison kernel changes:
>
> $$
> |\sigma(\Delta) - \mathbf{1}(\Delta > 0)| \leq e^{-|\Delta|}.
> $$
>
> Thus, the induced surrogate gap is controlled by $\mathbb{E}[e^{-|\Delta|}]$, and becomes small when chosen/rejected score distributions are well separated with limited mass near the decision boundary.
>
> For **IRPM–Mean/Median/Interval**, analogous bounds are possible in principle, but they additionally depend on (i) concentration of the opponent-group statistic, and (ii) how much score mass lies near the threshold/margin region. Because such finite-$G$ bounds are necessarily assumption-dependent, we did not state a universal optimality theorem in the current submission.
> ## W3: Wall-clock training and inference time comparisons
> We compare the actual training and inference time of IRPM during RL with two representative baselines: the B-T model and the pairwise GRM (RM-R1). For detailed experimental settings, results, and analyses, please refer to our response to Reviewer T8wH Weakness1, which could not be included here due to char limitations.

---

> > ### Author Rebuttal · Reviewer_uugc · 2026-04-01
> >
> > I thank the reviewers for their response which has addressed my questions and concerns.
> > I will consider updating the score.

---

> > > ### Author Response · Authors · 2026-04-03
> > >
> > > Thank you for your response and for your consideration. We are glad to know that our response has addressed your questions and concerns. We sincerely appreciate your time and valuable feedback, which have helped improve the paper.

---

### Official Review · Reviewer_j1rJ · 2026-03-07

**Soundness:** 3
**Presentation:** 3
**Significance:** 2
**Originality:** 2
**Overall Recommendation:** 4
**Confidence:** 3

**Summary:**

The authors propose IRPM, a method for training pointwise generative reward models (GRMs) from pairwise preference data. Given a preference tuple $(x, y_c, y_r)$ consisting of the prompt $x$, a chosen response $y_c$, and a rejected response $y_r$, the GRM generates critiques $c_i^c$ and scalar scores $s_i^c$ for the pair $(x, y_c)$, and analogously $c_j^r, s_j^r$ for $(x, y_r)$. Multiple samples are drawn for each response via a process the authors call _intergroup sampling_. The resulting groups of scores are compared across the chosen and rejected responses to construct rollout-level rewards derived from a Monte Carlo approximation of the Bradley-Terry preference probability. These rewards are then used to optimize the GRM via a reinforcement learning objective (GRPO). Experiments show that fine-tuning Qwen3 models of various sizes using IRPM improves performance over the base models across several reward-model benchmarks.

**Compliance With Llm Reviewing Policy:**

Affirmed.

**Final Justification:**

The authors' rebuttal addressed my methodological concerns adequately. I reflected this by increasing the score to 4.

**Key Questions For Authors:**

- **Q1:** How sensitive is IRPM to the choice of the underlying GRM backbone? In the experiments, intergroup sampling and reward generation are performed using the same model that is subsequently fine-tuned. It would be helpful to understand whether the method remains effective when the scores and critiques are generated by a different model (e.g., a weaker or stronger LLM judge).
-  **Q2:** The main motivation for IRPM is the reduction of reward evaluation complexity from $\mathcal{O}(N^2)$ to $\mathcal{O}(N)$. However, it is unclear whether this reduction is practically important in typical RLHF settings, where the number of candidate responses per prompt is often small (e.g., 4-8). Since the ultimate objective is to obtain a high-quality reward signal rather than to minimize comparisons, it would be helpful to compare IRPM against pairwise GRM methods under an equal compute budget. For example, if both approaches are allowed the same number of reward evaluations or comparisons, does IRPM still outperform exhaustive pairwise methods, or does the Monte Carlo approximation introduce limitations in reward quality?

**Limitations:**

yes

**Strengths And Weaknesses:**

**Strengths:**
- Compared to approaches that rely on exhaustive pairwise comparisons between candidate responses, the proposed method reduces the computational cost of reward evaluation from $\mathcal{O}(N^2)$ to $\mathcal{O}(N)$.
- The proposed training objective and reward design variants are clearly motivated and supported with ablation studies.
- The paper is well-presented

**Weaknesses:**
- Since the reward signal is derived from scores generated by the GRM itself, the effectiveness of IRPM depends strongly on the quality of the underlying LLM judge. However, the experiments only perform intergroup sampling using the same model that is being fine-tuned. This raises the question of whether the improvements stem from the proposed training procedure or from properties of the specific backbone model used.
- It is unclear how the proposed reward generation scheme compares in informativeness to full pairwise GRM approaches under an equal compute budget.

---

> ### Author Rebuttal · Authors · 2026-03-31
>
> Dear Reviewer j1rJ,
>
> Thank you for recognizing the computational efficiency of IRPM. Below are our responses to the weaknesses and questions you raised.
> ## W1 & Q1: Sensitivity to the GRM Backbone
> We conduct additional experiments using Llama3.1-8B-Instruct as the backbone. This model differs significantly in architecture and training data from the Qwen3. The results are shown below. The "+" sign indicates the improvement over the backbone.
>
> | Backbone | RM-Bench | JudgeBench | RewardBench | PPE | Overall  |
> |--|--|--|--|--|--|
> | Llama3.1-8B-Instruct (pairwise) | 54.0 | 32.3  | 69.5 | 55.6 | 52.9 |
> | J1-Llama-8B (pairwise) | 73.4 | 42.0  | 85.7 | 59.8 | 65.2 **+12.3** |
> | Llama3.1-8B-Instruct (pointwise) | 39.7 | 37.7  | 50.7 | 31.8 | 40.0 |
> | IRPM-Llama3.1-8B (pointwise) | 65.7 | 57.7  | 78.1 | 57.6 | 64.8 **+24.8** |
>
> IRPM outperforms the pointwise backbone by **+24.8** and achieves performance close to that of the pairwise GRM J1-Llama-8B. These findings strongly suggest that **the effectiveness of IRPM arises from its training paradigm itself**, rather than from the properties of any specific backbone model, thereby demonstrating strong generalization ability and robustness.
>
> We hypothesize that IRPM is relatively robust to the choice of backbone for two main reasons:
> - **Prompt-conditioned comparability**. IRPM performs intergroup comparisons under a shared prompt-conditioned reference (Section 6), which makes optimization less sensitive to the absolute score scale of a particular model.
> - **Distribution matching**: The Monte Carlo Bradley–Terry objective (Eq. 4) encourages the score distribution of chosen responses to dominate that of rejected responses. Since this objective is defined at the distributional level, it is not tied to any specific model architecture and should generalize across different GRM backbones.
> ## W2: Comparison Under Equal Compute Budget
> To address this concern, we conduct additional experiments in which IRPM and pairwise GRMs are compared **under the same total compute budget**, measured by the total number of reward evaluations.
>
> Specifically, for $n = 8$ candidate responses:
> - **Pairwise GRM**: we keep $8\times 4 = 32$ comparisons, which is the same as in the post-training setup of RRM in Sec 5.
> - **IRPM**: we sample 4 rollouts per response and apply voting@4, which also corresponds to the same total comparison budget.
>
> We then use these reward models for the post-training of Qwen2.5-7B and evaluate the final policy's performance on MMLU-PRO. We also include RM-R1 as a strong baseline.
>
> | Models | Comparisons | Performance (MMLU_PRO) |
> |-|-|-|
> | RRM     | 32|50.9%|
> | RM-R1   | 32|54.8%|
> | IRPM-32B| 8 | 54% |
> | IRPM-32B| 32|56.6%|
>
> The results show that when IRPM uses the same compute budget as the pairwise methods (32 evaluations), the resulting policy achieves a performance of 56.6%, **outperforming both RM-R1 and RRM**. Moreover, as the number of IRPM evaluations increases from 8 to 32, the policy performance further improves from 54.0% to 56.6%.
>
> **Does the Monte Carlo approximation introduce limitations in reward quality?** As stated in Section 6, this approach of intergroup comparisons and rollout-level reward decomposition actually produces smoother, lower-variance reward signals. This stabilizes the optimization process, improving both training efficiency and final performance. In contrast, while traditional pairwise methods are information-complete, they can generate sparser signals with higher variance.
>
> ## Q2: Practical Importance of O(n) Reduction
> We validate the practical significance of reducing the complexity from $O(n^2)$ to $O(n)$ through the following theoretical analysis and experimental results.
>
> As shown below, this advantage becomes increasingly significant as the candidate set grows.
> | Candidates (n) | Pairwise O(n²) | IRPM O(n) | Reduction |
> |-|-|-|-|
> |4|6|4|1.5×|
> |8|28|8|3.5×|
> |16|120|16|7.5×|
>
> For a typical case with $n = 8$ candidates, IRPM reduces the number of reward comparisons from 28 to 8, corresponding to a 3.5× reduction in reward computation. We have also added a comparison of the actual post-training time: IRPM takes **47h vs. 108h**. (For details, please refer to our response to Reviewer T8wH, W1.)
>
> Furthermore, as the demand for more extensive exploration in RLHF grows, scenarios involving a large number of candidates (e.g., n > 16) from techniques like beam search or large-scale Best-of-N sampling will become more common. In such cases, IRPM's $O(n)$ complexity will be increasingly advantageous compared to pairwise GRMs.
>
> We sincerely appreciate your helpful feedback and welcome further discussion on these additions.

---

> > ### Author Rebuttal · Reviewer_j1rJ · 2026-04-01
> >
> > Thank you for the detailed rebuttal and additional experiments. The new results with Llama3.1-8B-Instruct as backbone (W1/Q1) and the equal compute budget comparison (W2) adequately address my main methodological concerns. I am updating my score from 3 → 4.

---

> > > ### Author Response · Authors · 2026-04-02
> > >
> > > Thank you for your response and for confirming that your concerns have been resolved. Your feedback has been invaluable in helping us improve the paper. We sincerely appreciate the time and effort you took to review our rebuttal and update your score.

---

### Official Review · Reviewer_T8wH · 2026-03-10

**Soundness:** 3
**Presentation:** 3
**Significance:** 3
**Originality:** 3
**Overall Recommendation:** 4
**Confidence:** 3

**Summary:**

The paper proposes Intergroup Relative Preference Modeling (IRPM), a method to train pointwise generative reward models using pairwise preference data. It performs intergroup comparisons between chosen and rejected response groups to estimate pointwise rewards, reducing the computational cost from O(n²) to O(n) in RLHF training. Experiments show that IRPM achieves state-of-the-art performance among pointwise reward models while maintaining competitive accuracy with pairwise models and requiring less computation.

**Compliance With Llm Reviewing Policy:**

Affirmed.

**Final Justification:**

The paper introduces Intergroup Relative Preference Modeling (IRPM), a method for training pointwise generative reward models from pairwise preference data. By performing intergroup comparisons between sets of chosen and rejected responses, the approach estimates pointwise rewards while reducing computational complexity from O(n^2) to O(n) in RLHF training. Experimental results demonstrate that IRPM achieves state-of-the-art performance among pointwise reward models, while remaining competitive with pairwise approaches and requiring substantially less computation.

During the rebuttal phase, the authors have satisfactorily addressed my concerns. I therefore maintain my original positive evaluation.

**Key Questions For Authors:**

Please refer to the “Strengths and Weaknesses” section. I would be willing to reconsider and potentially increase the rating if the authors adequately address the concerns I raised.

**Limitations:**

This point is not discussed in the manuscript. Relevant suggestions can be found in the “Strengths and Weaknesses” section.

**Strengths And Weaknesses:**

Strength:
1. The paper introduces IRPM, which extends the Bradley–Terry preference framework to intergroup comparisons and enables training pointwise generative reward models directly from pairwise preference data. This provides a new bridge between pairwise supervision and pointwise reward modeling.
2. The method addresses the computational bottleneck of pairwise reward modeling by reducing reward evaluation complexity from O(n^2) to O(n). This practical design makes the approach more scalable for RLHF training.
3. The paper presents the motivation, theoretical derivation, and algorithmic pipeline in a logical sequence, and includes figures and ablation studies that clearly explain the design choices and empirical behavior of the method.

Weakness:
1. The paper lacks a corresponding time-efficiency analysis. Under the same dataset and training conditions, it would be helpful to compare the training time and computational resource consumption (e.g., hardware usage) of IRPM with methods such as RM-R1, standard generative reward models, and discriminative reward models. In addition, the paper should also report the time cost when these reward models are used during inference or downstream training (e.g., RL), to provide a clearer comparison of practical efficiency.
2. In terms of baselines, it would be beneficial to include some strong closed-source models, especially recent ones such as GPT-5 and Claude Sonnet 4. For example, these models could be prompted to perform pairwise or pointwise preference judgments directly. Such comparisons would help better position the proposed method against current state-of-the-art systems.
3. The conclusions in the Different Reward Design Methods section could benefit from deeper analysis. Currently, the discussion appears somewhat descriptive and lacks clear insights into why certain reward designs perform better than others. Providing more detailed explanations of the underlying mechanisms or training dynamics would strengthen the section and make the findings more informative.
4. It would also be helpful to include test-time scaling curves, showing how performance changes as the number of samples or votes increases during inference. This could be presented briefly, but it would make the impact of inference-time scaling clearer.

---

> ### Author Rebuttal · Authors · 2026-03-31
>
> Dear Reviewer T8wH,
>
> Thank you for recognizing the efficiency and practical value of IRPM. Below are our responses to the weaknesses and questions you raised.
> ## W1: Time-Efficiency and Computational Cost
> We compare IRPM with two representative baselines: the B-T model and the pairwise GRM (RM-R1).
>
> For fairness, in the RM training stage, all methods use the same backbone Qwen3-32B, dataset and hardware (8× AMD MI300X GPUs). In the post-training, we use the same RL data and configuration, with 8× AMD MI300X GPUs for training and another 8× AMD MI300X GPUs for RM serving.
> |Models|TrainingTime(h)|RL Time(h)|RM Inference Time(s/step)|
> |-|-|-|-|
> |B-T|21|38|133|
> |Pairwise GRM|60|108|560|
> |IRPM|82|47|210|
>
> The results show two expected trade-off.
> - B-T is fastest because it only predicts a scalar reward, but it lacks interpretability and inference-time scalability.
> - Compared with pairwise GRM, although IRPM incurs 22h (82h vs. 60h) additional RM training cost, it **saves 61h (47h vs. 108h)** in the subsequent RL, yielding a net end-to-end reduction. This comes from reducing RM inference time from **560s/step to 210s/step**.
>
> ## W2: Comparison with Strong Closed-Source Models
> We add results with strong closed-source judges, including GPT-5 and Claude-4-Sonnet, under pairwise and pointwise settings. For pointwise evaluation, we use the same prompt as IRPM; for pairwise evaluation, we use the Arena-Hard prompt.
> |Models|RM-Bench|JudgeBench|RewardBench|PPE|Overall|
> |-|-|-|-|-|-|
> |GPT-4o (pairwise)|79.5|68.9|89.1|67.4|76.2|
> |Claude-4-Sonnet (pairwise)|88.1|82.6|95.4|70.1|84.1|
> |GPT-5 (pairwise)|89.7|88.6|91.8|76.5|86.7|
> |GPT-5 (pointwise)|88.1|89.4|90.4|74.8|85.7|
> |IRPM-32B-Mean (pointwise)|79.6|74.6|87.3|67.2|77.2|
> |voting@16|85.2|79.0|91.0|71.8|81.8|
>
> Although IRPM still trails the strong closed-source pairwise judges, likely due to their larger scale, IRPM-32B-Mean outperforms GPT-4o (pairwise) overall and IRPM-32B-Mean (voting@16) surpasses GPT-5 (pointwise) on RewardBench while achieving performance close to it on RM-Bench and PPE.
>
> ## W3: Deeper Analysis of Reward Design Methods
> We expand the analysis of reward design variants. Their performance ranks as:
>
> IRPM-Preference < IRPM-Interval < IRPM-AUC < IRPM-Median ≈ IRPM-Mean.
>
> We strengthen the analysis from three aspects: **compatibility with GRPO optimization, robustness of the reward signal, and reward sparsity**.
> - Although IRPM-Preference is the closest to the Monte Carlo B-T objective, it directly uses preference strength as reward and keeps encouraging larger chosen–rejected score gaps. This is poorly matched to GRPO, whose update is based on within-group normalization. The resulting positive feedback enlarges intra-group score variance and eventually causes instability. This is reflected in Figure 3 (right), where IRPM-Preference shows steadily increasing variance and training collapse.
> - IRPM-AUC replaces soft preference strength with a hard ranking signal, which bounds the reward and improves stability. However, it still compares each rollout against individual opponent samples, making it more sensitive to sampling noise than group-level aggregate rewards. Consistently, its score variance is the second highest in Figure 3.
> - IRPM-Interval uses a conservative confidence-interval criterion and only gives positive reward when margin separation remains valid under uncertainty. This improves robustness, but makes the reward much sparser. We quantified this by tracking the average training reward:
> |Models|step0|step200|step400|step600|step800|
> |-|-|-|-|-|-|
> |IRPM-Preference|0.639|0.689|0.692|0.699|0.708|
> |IRPM-AUC|0.624|0.714|0.731|0.746|0.762|
> |IRPM-Interval|**-0.224**|0.012|0.036|0.112|**0.171**|
> |IRPM-Median|0.333|0.480|0.505|0.535|0.559|
> |IRPM-Mean|0.323|0.503|0.526|0.555|0.583|
>
> The substantially lower reward magnitude of IRPM-Interval indicates much sparser effective learning signals.
> - Mean and Median are better aligned with GRPO because they do not continuously amplify margins. Instead, they compare each rollout against an aggregate statistic of the opposite group, which is both more robust to noisy samples and denser than Interval-style rewards. As shown in Figure 3 (right), they have the smallest score variance, indicating greater training stability. This explains their consistently strongest performance.
>
> ## W4: Test-Time Scaling Curves
> We add more test-time scaling results beyond the voting@2 and voting@8 numbers originally reported in §4.2.
> |voting@n|RM-Bench|JudgeBench|RewardBench|PPE|Overall|
> |-|-|-|-|-|-|
> |IRPM-Mean-32B|79.6|74.6|87.3|67.2|77.2|
> |voting@2|82.5|75.1|89.3|70.1|79.3 **+2.1**|
> |voting@4|84.2|74.9|90.2|70.1|79.9 **+2.7**|
> |voting@8|84.8|77.7|91.2|71.5|81.3 **+4.1**|
> |voting@16|85.2|79.0|91.0|71.8|81.8 **+4.6**|
>
> These results show a clear overall inference-time scaling trend. As discussed in §4.2, IRPM benefits substantially from voting, which is one of the practical advantages of pointwise GRMs.

---

> > ### Author Rebuttal · Reviewer_T8wH · 2026-04-01
> >
> > Thank you for your clarification. I would like to maintain my original rating.

---

> > > ### Author Response · Authors · 2026-04-02
> > >
> > > Thank you for your response and for confirming that our clarification has addressed your concerns. We would be grateful if this could be taken into consideration in your final assessment. We sincerely appreciate your time and thoughtful feedback, which have helped improve the paper.

---

### Official Review · Reviewer_QGQv · 2026-03-13

**Soundness:** 3
**Presentation:** 3
**Significance:** 3
**Originality:** 3
**Overall Recommendation:** 4
**Confidence:** 3

**Summary:**

This paper addresses the issue that pointwise reward models often underperform while pairwise reward models require prohibitively large amounts of data. It proposes a method called Intergroup Relative Preference Modeling (IRPM), which transforms point-to-point comparisons into point-to-group comparisons. By fully leveraging the sampling diversity of generative reward models, IRPM enriches reward signals while reducing comparison complexity (from O(n²) to O(n)). Experiments on multiple authoritative benchmarks, such as JudgeBench and RewardBench, demonstrate that IRPM matches or surpasses the performance of state-of-the-art reward models trained on hundreds of thousands of data points. Thus, IRPM offers a cost-effective and efficient approach to training reward models.

**Compliance With Llm Reviewing Policy:**

Affirmed.

**Final Justification:**

This paper proposes Intergroup Relative Preference Modeling (IRPM), transforming point-to-point into point-to-group comparisons. It reduces the reward evaluation complexity in RLHF from O(n²) to O(n), significantly enhancing computational and data efficiency for learning from limited data.

The rebuttal resolved my concerns. The added matched-budget comparison shows a slight performance gap versus traditional Pairwise GRM, clarifying the efficiency-performance trade-off. Additional baselines on the 21K subset and temperature ablations further support the claims on data efficiency and training stability.

Overall, the rebuttal confirms my initial assessment, and I maintain my score of 4 (Weak accept).

**Key Questions For Authors:**

See the ‘Weaknesses’.

**Limitations:**

yes

**Strengths And Weaknesses:**

**Strengths**:
1. Reduced computational complexity: The complexity of reward computation is reduced from O(n²) to O(n).
2. High data efficiency: Using only 21K data points, IRPM achieves performance comparable to that of reward models trained on much larger datasets.
3. Comprehensive and convincing experiments: The performance improvements of IRPM are validated across multiple authoritative reward modeling benchmarks. Ablation studies further examine the impact of group size G, Chain-of-Thought (CoT) reasoning, data filtering, and different reward design choices (e.g., Mean, Median).

**Weaknesses**:
1. Although the computational complexity is reduced from O(n²) to O(n), the actual number of samples required by IRPM is 2G (with separate sampling for the chosen and rejected responses) when the nominal sample size is G. This implies that as G increases, there exists a crossover point in computational cost between traditional methods and IRPM. Are there experiments demonstrating the trade-off between efficiency and performance for traditional methods versus IRPM as G varies from small to large? It is important to evaluate this, given the non-negligible additional overhead introduced by IRPM itself.

2. Regarding the attribution of data efficiency, could the authors verify, on the exact same 21K HelpSteer3-Preference dataset, whether IRPM significantly outperforms (1) a standard pointwise reward model (trained with MSE loss) and (2) a standard pairwise reward model (trained with, e.g., Pairwise Ranking Loss)?

3. The repeated GRM sampling cleverly utilizes the output diversity of the generative model to obtain more unbiased intra-group reward preference information. I am curious about the impact of different sampling temperatures on the effectiveness of IRPM.

---

> ### Author Rebuttal · Authors · 2026-03-31
>
> Dear Reviewer QGQv,
>
> Thank you for recognizing IRPM as a cost-effective and efficient approach to reward modeling. Below are our responses to the weaknesses and questions you raised.
> ## W1: 2G sampling overhead and the efficiency–performance trade-off
> When the nominal group size is G, IRPM indeed samples separately from the chosen and rejected, resulting in 2G total rollouts. We would like to clarify two aspects:
> - The complexity reduction claimed in the paper mainly refers to the **reward evaluation/comparison** cost during RL training, rather than rollout generation. For a traditional pairwise GRM, converting relative preferences into scalar rewards over a candidate set of size $n$ typically requires all-pairs comparisons, leading to $O(n^2)$ cost. In contrast, IRPM only requires pointwise scoring for each candidate, giving $O(n)$ reward evaluation cost.
> - The extra overhead of IRPM mainly comes from **training-time rollout sampling**. The paper includes a group-size ablation (Sec4.3): increasing G from 2 to 4 brings clear gains, while increasing from 4 to 8 shows diminishing returns, indicating that the best operating point is at a moderate G, rather than a large one. In fact, the current results suggest that G=4 provides the best performance–computation trade-off.
>
> To address this concern more directly, we additionally compare IRPM and pairwise methods under the **same computation budget**. Using Qwen3-8B for all models, when $G=2$, the pairwise GRM samples 4 rollouts, while IRPM samples 2 rollouts for chosen and 2 for rejected. Evaluation follows Sec4.3, using the average of RM-Bench and JudgeBench:
>
> |G|Pairwise GRM (2G)|IRPM(G)|
> |-|-|-|
> |2|73.5|71.6|
> |4|74.9|74.0|
> |8|75.1|73.7|
>
> These results show that under matched budgets, IRPM remains competitive, while retaining the key advantage of linear-time reward evaluation at RL time.
> In addition, we measured the actual wall-clock training and inference time, please refer to our response to Reviewer T8wH W1 (Due to character limits, it cannot be displayed here).
> ## W2: Comparison with standard pointwise/pairwise RMs on the same 21K data
> On the same 21K HelpSteer3 subset, we train the following baselines:
> - **Pointwise RM with MSE Loss.** Since HelpSteer3 only provides pairwise preferences rather than absolute scalar scores, a vanilla pointwise MSE model is not directly applicable. To enable this comparison, we convert each preference pair into two pointwise examples with symmetric pseudo-labels, $y_c=1,y_r=0$, and train an MSE-based pointwise model on the same 21K subset.
> - **Pairwise RM with Ranking Loss.** The B–T reward model is trained with pairwise ranking loss and producing pointwise scalar scores. In Table 2 of the paper, BT-Qwen3-32B is exactly this baseline trained on the same 21K data.
> - Pairwise GRM.
>
> |Model|Training data|RM-Bench|JudgeBench|Overall|
> |-|-|-|-|-|
> |Pointwise RM (MSE)|21K|70.7|67.1|68.9|
> |Pairwise RM (BT)|21K|74.3|68.9|71.6|
> |Pairwise GRM|21K|84.3|71.1|77.7|
> |IRPM-Mean-32B|21K|79.8|74.6|77.2|
>
> These results show that, under the same 21K training data, IRPM substantially outperforms a pointwise RM (MSE) by **+8.3** and a pairwise RM (B-T) by **+5.6**, while approaching the performance of the pairwise GRM.
>
> **We note that Table 2 contains a typographical error: "RWBench" should be "RM-Bench". We will correct it in the revised version.**
>
> ## W3: Temperature Impact on Repeated GRM Sampling
> We conduct an additional ablation on the sampling temperature used in repeated GRM sampling:
> |Temperature|RM-Bench|JudgeBench|Overall|
> |-|-|-|-|
> |0.3|76.0|68.1|72.1|
> |0.7|77.0|68.0|72.5|
> |1.0|77.9|70.0|**74.0**|
> |1.5|77.8|68.4|73.1|
>
> Overall, IRPM performs reasonably well across a broad temperature range, which suggests that the method is fairly robust to this hyperparameter. Among the tested values, temperature = 1.0 gives the best overall performance.
>
> We also examine the average training reward dynamics:
> |Temperature|step0|step100|step200|step300|step400|step500|step600|step700|step800|
> |-|-|-|-|-|-|-|-|-|-|
> |0.3|0.371|0.264|0.311|0.323|0.329|0.339|0.349|0.351|0.344|
> |0.7|0.294|0.262|0.309|0.332|0.339|0.358|0.375|0.38|0.379|
> |1.0|0.316|0.244|0.307|0.324|0.330|0.350|0.367|0.379|0.376|
> |1.5|**-0.625**|**-0.05**|0.15|0.231|0.278|0.318|0.347|0.358|0.369|
>
> These observations are consistent with the intuition behind IRPM. If the temperature is too low, within-group samples become overly similar, reducing the diversity needed for informative intergroup comparisons. If it is too high, the sampled scores become noisier and optimization is less stable, especially early in training. The behavior at temperature 1.5 is particularly illustrative: the average reward starts negative but rises quickly, suggesting stronger early exploration but also higher noise.
>
> Overall, **a moderate temperature balances diversity and stability**. We use temperature = 1.0 as the default because it is a stable and conservative choice across settings.

---

> > ### Author Rebuttal · Reviewer_QGQv · 2026-04-03
> >
> > Thank you for your response. I appreciate the clarifications provided, which have fully addressed my questions. I am satisfied with the authors' rebuttal and will keep my original score unchanged.

---

> > > ### Author Response · Authors · 2026-04-03
> > >
> > > Thank you for your positive feedback. We are pleased to know that our clarifications have addressed your concerns and that you are satisfied with our response. We sincerely appreciate your valuable comments and suggestions, which have helped us improve the paper.

---

### Decision · Program_Chairs · 2026-04-30

**Decision:**

Accept (regular)

**Comment:**

This paper proposes an intergroup relative preference modeling framework for training pointwise generative reward models from pairwise preference data, with the goal of combining the scalability and flexibility of pointwise reward evaluation with the performance advantages of pairwise supervision.

Reviewers viewed the paper positively overall and highlighted the method’s practical significance, particularly its reduction of reward evaluation complexity from quadratic to linear time in downstream RL settings, its strong data efficiency, and its competitive performance across multiple reward-modeling benchmarks. The experimental section was generally regarded as comprehensive, with useful ablations on group size, reward design, and sampling behavior, and the paper’s core motivation and algorithmic development were found clear and well structured.

The main concerns focused on clarifying the true efficiency tradeoff, strengthening matched-budget and same-dataset baseline comparisons, and providing deeper analysis of sampling temperature, time efficiency, and reward design choices. These concerns were addressed well in the rebuttal. Overall, the paper makes a technically sound and practically meaningful contribution to reward modeling for RLHF, and it meets the bar for acceptance at ICML.